# Curriculum Design for Trajectory-Constrained Agent: Compressing Chain-of-Thought Tokens in LLMs

**Georgios Tzannetos**
MPI-SWS
gtzannet@mpi-sws.org

**Parameswaran Kamalaruban**
Featurespace Innovation Lab, Visa
kaparame@visa.com

**Adish Singla**
MPI-SWS
adishs@mpi-sws.org

## Abstract

Training agents to operate under strict constraints during deployment, such as limited resource budgets or stringent safety requirements, presents significant challenges, especially when these constraints render the task complex. In this work, we propose a curriculum learning strategy that gradually tightens constraints during training, enabling the agent to incrementally master the deployment requirements. Inspired by self-paced learning techniques in unconstrained reinforcement learning (RL), our approach facilitates a smoother transition to challenging environments by initially training on simplified versions of the constraints and progressively introducing the full deployment conditions. We provide a theoretical analysis using an RL agent in a binary-tree Markov Decision Process (MDP) to demonstrate that our curriculum strategy can accelerate training relative to a baseline approach that imposes the trajectory constraints from the outset. Moreover, we empirically validate the effectiveness and generality of our method across both RL and large language model (LLM) agents in diverse settings, including a binary-tree MDP, a multi-task navigation domain, and a math reasoning task with two benchmarks. These results highlight the potential of curriculum design in enhancing the efficiency and performance of agents operating under complex trajectory constraints during deployment. Moreover, when applied to LLMs, our strategy enables compression of output chain-of-thought tokens, achieving a substantial inference speedup on consumer hardware, demonstrating its effectiveness for resource-constrained deployment.

## 1 Introduction

Recent advances in reinforcement learning (RL) have enabled agents to excel in diverse applications, including robotics, gaming, and large language models (LLMs) [1, 2, 3, 4, 5]. A common assumption in these domains is that the resources available during training, such as computational power and time, remain equally accessible during deployment. However, in many real-world scenarios, deployment-time constraints such as limited resource budgets or strict safety requirements significantly diverge from the training environment, posing unique challenges that standard RL methods often overlook.

Consider a robot navigating from point A to B that is expected to reach its destination, while ensuring that travel time and energy consumption do not exceed predefined limits. Similarly, consider an LLM being used for answering a mathematical question, where a user requires that the response arrive at the correct answer while using no more than a specified number of chain-of-thought (CoT) tokens. Recent works have studied how to control the length of LLM responses at test-time via training objectives or prompting, leading to improved reasoning capabilities [6, 7, 8, 9]. We consider it from the prospective of designing a curriculum strategy that can enable LLMs to output responses more desirable than the initial response in terms of reduced inference time and computational cost, while preserving the answer's accuracy (Figure 1). Motivated by these examples, we consider the

39th Conference on Neural Information Processing Systems (NeurIPS 2025).

> **Test-set Math Question**: Below is an instruction that describes a task. Write a response that appropriately completes the request. ### Instruction: John adopts a dog. He takes the dog to the groomer, which costs $100. The groomer offers him a 30% discount for being a new customer. How much does the grooming cost? ### Response: Let's think step by step.

> **Base Model Response**: The grooming cost is $100. The groomer offers a 30% discount, which means John will pay 100 - (30% of 100) = $70 for the grooming. So, the grooming cost is $70. #### 70
> The answer is: 70

> **Fine-tuned Model Response (50K Episodes)**: The grooming cost is $100. The groomer offers a 30% discount, so the discount amount is 30% of $100, which is 0.3 * $100 = $30. Therefore, the final cost of the grooming is $100 - $30 = $70. #### 70
> The answer is: 70

> **Fine-tuned Model Response (100K Episodes)**: The original cost of grooming is $100. The discount is 30% of $100, which is $100 * 0.30 = $30. The final cost is $100 - $30 = $70. #### 70
> The answer is: 70

> **Fine-tuned Model Response (200K Episodes)**: Discount=0.3*100=30 Cost=100-30=70 #### 70
> The answer is: 70

> **Fine-tuned Model Response (400K Episodes)**: 100*.7=70 #### 70
> The answer is: 70

Figure 1: Inference on an unseen GSM8K test-set question using different model checkpoints obtained during fine-tuning. These responses showcase that the fine-tuned models progressively learn to generate shorter yet correct solutions by implicitly performing intermediate reasoning steps.

following research question: *How can we effectively guide the training process when deployment-time constraints are known in advance, in order to achieve the best performance under those constraints?*

A natural approach is to impose deployment-time constraints during training. However, doing so exacerbates the sparse reward problem common in RL, since the agent only receives a reward upon both task completion and strict constraint adherence. This setup contrasts with standard constrained RL formulations [10, 11], where the goal is to maximize the expected reward subject to a bounded expected cost, a relaxation that does not fully capture the strict nature of deployment requirements. Alternatively, curriculum learning methods developed for contextual RL [12, 13, 14] can treat the permissible cost budget as an input context and gradually shift towards the deployment budget. Yet, these methods incur significant computational overhead because they require extensive performance evaluations across the entire high-dimensional context space, an issue that becomes particularly problematic in domains such as LLMs, where each rollout can be costly.

In this work, we propose a novel curriculum strategy that adaptively adjusts the permissible cost budget during training based on the agent's current performance. This strategy starts with relaxed trajectory constraints and adaptively tightens them, facilitating a smoother transition to stringent deployment conditions. Our main results and contributions are as follows:

1. We introduce a computationally-efficient curriculum strategy tailored to deployment-time constraints in RL.

2. We provide a theoretical analysis on a binary-tree MDP to demonstrate that our adaptive curriculum strategy accelerates training relative to approaches that enforce strict deployment constraints from the beginning.

3. We validate our method's effectiveness and versatility through comprehensive experiments with both RL and LLM agents across multiple domains. The results highlight its potential to enhance agent performance under real-world deployment constraints, demonstrated by compressing LLM's output chain-of-thought tokens in math reasoning.[1]

## 1.1 Related Work

**Constrained Reinforcement Learning.** Constrained reinforcement learning (CRL) is typically formulated as a constrained Markov decision process (CMDP) [15], where the goal is to maximize the

---

[1]Github: https://github.com/machine-teaching-group/neurips2025-curriculum-llm-tokens

expected cumulative reward while satisfying expectation-based cost constraints. Standard approaches include Lagrangian methods [11, 16], which optimize a weighted combination of rewards and costs using a scalar multiplier, and trust-region methods [10, 17, 18] that enforce constraint satisfaction during policy updates. In these formulations, the objective is generally given by

$$\max_{\pi} \ \mathbb{E}_{\pi}\left[\sum_{\tau} \gamma^{\tau} \cdot R(s^{(\tau)}, a^{(\tau)})\right] \quad \text{subject to} \quad \mathbb{E}_{\pi}\left[\sum_{\tau} \gamma^{\tau} \cdot C(s^{(\tau)}, a^{(\tau)})\right] \leq \alpha,$$

where $R$ is the reward function, $C$ is the cost function, and $\alpha$ is the cost threshold. In contrast, our formulation enforces strict, trajectory-level constraint satisfaction by considering the problem

$$\max_{\pi} \ \mathbb{E}_{\pi}\left[\sum_{\tau} \gamma^{\tau} \cdot R(s^{(\tau)}, a^{(\tau)}) \cdot \mathbf{1}\left[\sum_{\tau} \gamma^{\tau} \cdot C(s^{(\tau)}, a^{(\tau)}) \leq \alpha\right]\right], \tag{1}$$

where the indicator function ensures that rewards are only granted if the entire episode adheres to the cost constraint. This stricter requirement inherently induces a sparse reward setting, posing unique challenges that are not addressed by conventional CRL methods.

**Curriculum for Contextual Reinforcement Learning.** Curriculum learning has been widely used in contextual RL to gradually expose agents to increasingly challenging tasks [19, 20]. Self-paced learning approaches [21, 22], such as SPDL [12, 23], SPACE [24], and CURROT [13], dynamically adjust the task distribution based on agent performance. Similarly, unsupervised environment design (UED) methods [25, 26, 27, 28] evolve the environment alongside the agent. ZPD-based strategies [29, 30, 31, 32], such as GOALGAN [33], and PROCURL-TARGET [14], select tasks of intermediate difficulty to maximize learning. While our setting could be framed as contextual RL, treating the cost budget as context and the deployment-time constraint as the target, this would require learning and evaluating a contextual policy across the full context space, which is impractical for domains like LLMs, where each (prompt, budget) pair demands separate rollouts. Instead, we train a non-contextual policy that adapts the cost budget per prompt, avoiding this computational overhead.

**Curriculum for Constrained Reinforcement Learning.** Another line of work has addressed curriculum strategies in constrained RL. A recent work [34] extends CURROT [13] to generate safe curricula that begin with low-cost tasks to reduce safety violations, then transition to high-reward contexts, and eventually converge to the target. Similarly, [35] introduces a human-inspired teaching strategy, where an artificial teacher guides the agent through progressively challenging, safety-preserving stages. However, both approaches assume expectation-based constraints, making them less applicable to our setting, which requires strict trajectory-level constraint satisfaction. Our work extends curriculum learning to this more stringent setting, offering both theoretical insights and empirical validation.

**Efficient LLM Reasoning with Compressed Output.** Recent works have begun exploring ways to improve the efficiency of LLM reasoning by controlling the model's output token length through prompting or training strategies [6, 7, 8, 9]. Fine-tuning LLMs to reduce verbosity or to adhere to length constraints specified in prompts has been studied, though primarily in the context of general-purpose text generation [36, 37]. For reasoning tasks, [7] proposed a simple RL method to train models to follow length constraints provided in the prompt, while [6] modified the RL objective to penalize longer correct responses. Similarly, [38] and [39] employed RL-based techniques to reduce LLM output length. However, these approaches do not consider strict user-specified test-time constraints; instead, they explore tradeoffs between accuracy and output length. In contrast, our work introduces a curriculum strategy that enables RL fine-tuning with the original sparse reward while ensuring that the resulting LLM strictly adheres to user-defined test-time constraints, without requiring constraint-specific prompts.

## 2 Formal Setup

**Multi-task RL.** We consider a multi-task reinforcement learning (RL) setting with a task or context space $\mathcal{X}$. Each task $x \in \mathcal{X}$ corresponds to a learning environment modeled as a contextual Markov Decision Process (MDP) $\mathcal{M}_x := \left(\mathcal{S}, \mathcal{A}, \gamma, \mathcal{T}_x, R_x, C_x, \alpha_x^*, P_x^0\right)$, where the state space $\mathcal{S}$, action space $\mathcal{A}$, and discount factor $\gamma$ are shared across all tasks, while the transition dynamics $\mathcal{T}_x : \mathcal{S} \times \mathcal{S} \times \mathcal{A} \to [0, 1]$, reward function $R_x : \mathcal{S} \times \mathcal{A} \to [0, 1]$, cost function $C_x : \mathcal{S} \times \mathcal{A} \to [0, 1]$, permissible cost budget $\alpha_x^*$, and initial state distribution $P_x^0 : \mathcal{S} \to [0, 1]$ are task-specific components [40, 41]. The collection of all environments is represented as $\mathcal{M} = \{\mathcal{M}_x : x \in \mathcal{X}\}$.

---

**Algorithm 1** Training RL Agents with Deployment-time Constraints

---
1: **Input:** RL agent's initial policy $\pi_1$
2: **for** $t = 1, 2, \ldots$ **do**
3:     Environment: *randomly* picks a task $x_t \in \mathcal{X}$.
4:     Teacher component: picks a training-time permissible cost budget $\alpha_t$.
5:     Student component: attempts the task $x_t$ via $K$ rollouts $\{\xi_i\}_{i=1}^K$ generated using the policy $\pi_t$ in the modified MDP $\widehat{\mathcal{M}}_{x_t}$.
6:     Student component: updates the policy to $\pi_{t+1}$ using the rollouts $\{\xi_i\}_{i=1}^K$ and the training-time reward function $\widehat{J}_{x_t}$.
7: **Output:** RL agent's final policy $\pi_{\text{end}} \leftarrow \pi_{t+1}$.

---

**RL agent and performance evaluation.** We consider an RL agent operating in an environment $\mathcal{M}_x \in \mathcal{M}$ using a contextual policy $\pi : \mathcal{S} \times \mathcal{X} \times \mathcal{A} \to [0, 1]$, which maps a state and task context to a probability distribution over actions. Given a task $x \in \mathcal{X}$, the agent attempts the task via a trajectory rollout obtained by executing its policy $\pi$ in the MDP $\mathcal{M}_x$. The trajectory rollout is denoted as $\xi = \left\{ (s^{(\tau)}, a^{(\tau)}) \right\}_{\tau=0,1,\ldots}$, where $s^{(0)} \sim P_x^0$. Let $\Xi$ denote the space of all possible trajectories. To evaluate the agent's performance, we define a trajectory-level reward function $\overline{J}_x : \Xi \to [0, 1]$ based on $R_x, C_x$, and the cost budget $\alpha_x^*$:

$$\overline{J}_x(\xi) := \left\{ \sum_\tau \gamma^\tau \cdot R_x(s^{(\tau)}, a^{(\tau)}) \right\} \cdot \mathbf{1}\left[ \sum_\tau \gamma^\tau \cdot C_x(s^{(\tau)}, a^{(\tau)}) \leq \alpha_x^* \right].$$

The agent's performance on task $x$ is then measured by the value function $V^\pi(x; \overline{J}_x) := \mathbb{E}_{\xi \sim \pi, \mathcal{M}_x}\left[ \overline{J}_x(\xi) \right]$. Finally, the uniform performance of the agent across all tasks in $\mathcal{X}$ is given by $\overline{V}^\pi := \mathbb{E}_{x \sim \text{Uniform}(\mathcal{X})}\left[ V^\pi(x; \overline{J}_x) \right]$.

**Training process of the RL agent.** During training, the agent employs a student component, responsible for policy updates, and a teacher component, which guides the student's learning process to find a policy that performs uniformly well across all tasks in $\mathcal{X}$, i.e., $\max_\pi V^\pi$. Training occurs in discrete steps indexed by $t = 1, 2, \ldots$, as formally described in Algorithm 1. At each step $t$, the environment *randomly* selects a task $x_t$ along with its deployment-time cost budget $\alpha_{x_t}^*$. However, the teacher component modifies this budget, replacing it with a training-time cost budget $\alpha_t$. This effectively transforms the original MDP $\mathcal{M}_{x_t}$ into a new MDP $\widehat{\mathcal{M}}_{x_t} := \left( \mathcal{S}, \mathcal{A}, \gamma, \mathcal{T}_{x_t}, R_{x_t}, C_{x_t}, \alpha_t, P_{x_t}^0 \right)$, thereby shaping the original trajectory-level reward function $\overline{J}_{x_t} = J_{x_t}^{\alpha_{x_t}^*}$ to obtain a new reward function $\widehat{J}_{x_t} = J_{x_t}^{\alpha_t}$, where:

$$J_x^\alpha(\xi) := \left\{ \sum_\tau \gamma^\tau \cdot R_x(s^{(\tau)}, a^{(\tau)}) \right\} \cdot \mathbf{1}\left[ \sum_\tau \gamma^\tau \cdot C_x(s^{(\tau)}, a^{(\tau)}) \leq \alpha \right].$$

The student component then attempts task $x_t$ by executing the policy $\pi_t$ for $K$ rollouts, denoted as $\{\xi_i\}_{i=1}^K$, within the modified MDP $\widehat{\mathcal{M}}_{x_t}$. After collecting these rollouts, the student component updates the policy to $\pi_{t+1}$ based on the current policy $\pi_t$, the selected task $x_t$, the teacher-shaped reward function $\widehat{J}_{x_t}$, and the set of rollouts $\{\xi_i\}_{i=1}^K$. Formally, the policy update is given by: $\pi_{t+1} \leftarrow L(\pi_t, x_t, \widehat{J}_{x_t}, \{\xi_i\}_{i=1}^K)$, where $L$ is a learning algorithm. Let $\pi_{\text{end}}$ denote the agent's final policy at the end of training. The *training objective* is to ensure that the performance of the policy $\pi_{\text{end}}$ is $\epsilon$-near-optimal, i.e., $(\max_\pi \overline{V}^\pi - \overline{V}^{\pi_{\text{end}}}) \leq \epsilon$. The primary objective of this work is to design a teacher component that achieves this training objective efficiently, both computationally and in terms of sample complexity.

## 3 Our Curriculum Strategy

In Section 3.1, we propose a curriculum strategy for selecting a permissible cost budget $\alpha_t$ (Algorithm 1 at Line 4). In Section 3.2, we provide a theoretical analysis in a binary-tree environment, demonstrating that the proposed strategy accelerates the agent's training process.

## 3.1 Curriculum Strategy

First, we discuss the challenges of using deployment-time cost budgets during training, i.e., selecting $\alpha_t = \alpha^*_{x_t}$ in Line 4 of Algorithm 1. When the reward function $R_x$ is goal-oriented (i.e., $R_x(s,a) = 1$ for all $a \in \mathcal{A}$ and $s \in \mathcal{G}_x$, and $R_x(s,a) = 0$ otherwise for some goal space $\mathcal{G}_x$), training with the deployment-time cost budget $\alpha^*_x$ can be highly ineffective, especially when $\alpha^*_x$ is small. In such cases, obtaining successful rollouts, i.e., trajectories $\xi$ where $\overline{J}_x(\xi) > 0$, using the randomly initialized policy $\pi_1$ is extremely difficult. As a result, directly using the original deployment-time budget $\alpha^*_x$ (or the corresponding reward function $\overline{J}_x$) during training may fail to provide a meaningful learning signal for policy updates, making the update rule $\pi_{t+1} \leftarrow L(\pi_t, x_t, \overline{J}_{x_t}, \{\xi_i\}_{i=1}^K)$ ineffective.

To address these challenges, we introduce a curriculum strategy that dynamically selects a permissible cost budget $\alpha_t$ based on the agent's current performance. At a high level, the intuition behind our approach is as follows: at each step $t$, given a task $x_t$, we shape the trajectory-level reward function $\overline{J}_{x_t}$ to obtain a new reward function $\widehat{J}_{x_t}$, which remains close to $\overline{J}_{x_t}$ while ensuring that the performance of the current policy $\pi_t$ on task $x_t$ is above a certain threshold. This naturally leads to a curriculum strategy for selecting $\alpha_t$.

**Formalization of the curriculum strategy.** At each step $t$, given a task $x_t$, the teacher component picks $\alpha_t$ by solving the following optimization problem for a given performance threshold $\beta > 0$:

$$\alpha_t \leftarrow \underset{\alpha \in \left[0, \frac{1}{1-\gamma}\right]}{\arg\min} \left(\alpha - \alpha^*_{x_t}\right)^2 \quad \text{subject to} \quad V^{\pi_t}(x_t; J^\alpha_{x_t}) \geq \min\{\beta, V^{\pi_t}(x_t; J^{\frac{1}{1-\gamma}}_{x_t})\}, \quad (2)$$

where $V^{\pi_t}(x_t; J^\alpha_{x_t}) = \mathbb{E}_{\xi \sim \pi_t, \mathcal{M}_{x_t}}\left[J^\alpha_{x_t}(\xi)\right]$ represents the value of policy $\pi_t$ on task $x_t$ under the reward function $J^\alpha_{x_t}$. Since $V^{\pi_t}(x_t; J^{\frac{1}{1-\gamma}}_{x_t})$ is the maximum achievable value due to the monotonically non-decreasing property of $V^\pi(x; J^\alpha_x)$ with respect to $\alpha$, the optimization problem in Eq. (2) always has at least one feasible solution: $\alpha = \frac{1}{1-\gamma}$. Our curriculum strategy adaptively selects $\alpha_t$ such that: (1) $\alpha_t$ remains as close as possible to the target permissible cost budget $\alpha^*_{x_t}$, and (2) the resulting reward function $J^{\alpha_t}_{x_t}$ provides sufficient learning signal for policy updates, as enforced by the constraint $V^{\pi_t}(x_t; J^{\alpha_t}_{x_t}) \geq \min\{\beta, V^{\pi_t}(x_t; J^{\frac{1}{1-\gamma}}_{x_t})\}$. Notably, we identify two particular cases of our strategy. First, when $\beta = 0$, the constraint is trivially satisfied for $\alpha = \alpha^*_{x_t}$, leading to $\alpha_t = \alpha^*_{x_t}$. This corresponds to the target curriculum, where the teacher directly picks the target $\alpha^*_{x_t}$. Second, when the maximum achievable value for the unconstrained problem $V^{\pi_t}(x_t; J^{\frac{1}{1-\gamma}}_{x_t})$ is zero, the curriculum selects the largest possible target parameter, aligning with the unconstrained curriculum.

**Practical implementation.** For any task $x$ and trajectory $\xi$, the reward function $J^\alpha_x(\xi)$ is monotonically non-decreasing in $\alpha$. Consequently, for any policy $\pi$, the value function $V^\pi(x; J^\alpha_x)$ is also monotonically non-decreasing with respect to $\alpha$. This property enables efficient solving of the optimization problem in Eq. (2) using binary search. The full procedure is presented in Algorithm 2.

---

**Algorithm 2** Teacher Component: Curriculum Strategy for Picking $\alpha_t$ (Line 4 of Algorithm 1)

---

1: **Input:** current policy $\pi_t$, current task $x_t$, and performance threshold $\beta$.
2: Generate a set of rollouts $\Xi = \{\xi_i\}_{i=1}^N$ using the policy $\pi_t$ in $\mathcal{M}_{x_t}$.
3: Set $\beta \leftarrow \min\left\{\beta, \frac{1}{N}\sum_{i=1}^N J^{\frac{1}{1-\gamma}}_{x_t}(\xi_i)\right\}$.
4: Binary search the smallest $\alpha_t \in \left[\alpha^*_{x_t}, \frac{1}{1-\gamma}\right]$ such that $\frac{1}{N}\sum_{i=1}^N J^{\alpha_t}_{x_t}(\xi_i) \geq \beta$.
5: **Output:** permissible cost budget $\alpha_t$.

---

## 3.2 Theoretical Analysis

We theoretically demonstrate the usefulness of our curriculum strategy in accelerating an agent's learning in a binary-tree environment. We select this basic RL setup, because it encapsulates the core challenge of our study, i.e., the sparse reward nature in strictly constrained RL problems.

**Binary-tree environment.** We consider a binary-tree MDP $\mathcal{M}_{\overline{x}} := (\mathcal{S}, \mathcal{A}, \gamma, \mathcal{T}_{\overline{x}}, R_{\overline{x}}, C_{\overline{x}}, \alpha^*_{\overline{x}}, P^0_{\overline{x}})$ of depth $H$ (focusing on the single-task setting $\mathcal{X} = \{\overline{x}\}$). The state space $\mathcal{S}$ consists of the nodes of

a binary tree, denoted as $\mathcal{S} = \{y_0^{(0)}, y_0^{(1)}, y_1^{(1)}, \cdots, y_0^{(H)}, y_1^{(H)}, \cdots, y_{2^H-1}^{(H)}\}$, where $y_i^{(h)}$ represents the $i$-th node (from the left) at level $h$. At each node, the agent can choose one of two actions, $\mathcal{A} = \{\text{LEFT}, \text{RIGHT}\}$, which deterministically transitions the agent to the respective child node in the tree. The initial-state distribution is concentrated at the root node $y_0^{(0)}$, with $P_{\overline{x}}^0(y_0^{(0)}) = 1$. Each trajectory through the tree terminates at a leaf node, representing a unique deterministic path from the root. All leaf nodes are designated as goal states, with rewards defined as $R_{\overline{x}}(s, a) = 1$ for all $s \in \{y_0^{(H)}, y_1^{(H)}, \cdots, y_{2^H-1}^{(H)}\}$ and $a \in \mathcal{A}$, and $R_{\overline{x}}(s, a) = 0$ otherwise. The cost function $C_{\overline{x}}$ is structured such that total trajectory cost increases monotonically from the leftmost to the rightmost leaf node. Denoting the total cost of a trajectory $y_0^{(0)}, \cdots, y_i^{(H)}$ as $C_{\overline{x}}(y_0^{(0)}, \cdots, y_i^{(H)})$, this condition is formally expressed as: $C_{\overline{x}}(y_0^{(0)}, \cdots, y_0^{(H)}) < C_{\overline{x}}(y_0^{(0)}, \cdots, y_1^{(H)}) < \cdots < C_{\overline{x}}(y_0^{(0)}, \cdots, y_{2^H-1}^{(H)})$. Any given tree structure can be rearranged to meet this property. A permissible cost budget, $\alpha_{\overline{x}}^* \in [0, 1]$, is specified as a constraint. Only the leftmost leaf node satisfies $C_{\overline{x}}(y_0^{(0)}, \cdots, y_0^{(H)}) \le \alpha_{\overline{x}}^*$, while all other leaf nodes exceed the budget, i.e., $C_{\overline{x}}(y_0^{(0)}, \cdots, y_i^{(H)}) > \alpha_{\overline{x}}^*$ for $i \in \{1, \ldots, 2^H - 1\}$. The agent begins with a uniformly random policy where $\pi_1(\text{LEFT} \mid y_i^{(h)}) = \pi_1(\text{RIGHT} \mid y_i^{(h)}) = 0.5, \forall i \in \{0, \ldots, 2^h - 1\}, h \in \{0, \ldots, H - 1\}$. The optimal policy is deterministic, always choosing LEFT at all relevant states, i.e., $\pi^*(\text{LEFT} \mid y_0^{(h)}) = 1.0, h \in \{0, \ldots, H - 1\}$.

Now, we compare the effects of two strategies in the binary-tree MDP described above:

1. **Target curriculum strategy:** The teacher always selects the target cost budget parameter $\alpha_{\overline{x}}^*$ (i.e., $\beta = 0$), meaning the learner receives a non-zero reward only upon completing a full successful trajectory (i.e., reaching the leftmost leaf). As we will demonstrate, this baseline strategy requires an exponential number of rollouts with respect to the tree depth $H$.

2. **Curriculum strategy with adaptive $\beta_t > 0$:** Here, the teacher gradually tightens the cost constraint by setting the performance threshold parameters to $\beta_t = 0.5 \cdot (1 - \epsilon)^{t-1}$ for $t = 1, \ldots, H$. This design ensures that a non-negligible fraction of rollouts yield a positive reward well before the optimal trajectory is reached, thereby promoting progressive learning.

The following theorem establishes that our curriculum strategy dramatically reduces the sample complexity compared to the baseline.

**Theorem 1.** *Consider the binary-tree MDP with depth $H$. Suppose the teacher employs the curriculum strategy as defined in Eq.* (2) *with threshold parameters $\beta_t = 0.5 \cdot (1 - \epsilon)^{t-1}$, for $t = 1, \ldots, H$, and selects the cost budget parameter $\alpha_t$ at each time step by solving*

$$\alpha_t \leftarrow \underset{\alpha \in \left[0, \frac{1}{1-\gamma}\right]}{\arg\min} \; (\alpha - \alpha_{\overline{x}}^*)^2 \quad \text{subject to} \quad V^{\pi_t}(\overline{x}; J_{\overline{x}}^\alpha) \geq \beta_t,$$

*where $\gamma$ is chosen so that $\gamma \geq 0.5^{\frac{1}{H}}$ (which guarantees that $\beta_t \leq \gamma^H$ for all $t$). Then, with probability at least $1 - \delta$, the learner obtains an $\epsilon$-suboptimal policy (i.e., $\pi_{H+1}(\text{LEFT} \mid y_0^{(h)}) \geq 1 - \epsilon, \forall h \in \{0, \ldots, H - 1\}$) after a total of $\sum_{t=1}^H K_t = \sum_{t=1}^H \frac{\ln\left(\frac{2}{\delta}\right)}{2 \cdot \epsilon^2 \cdot (1-\epsilon)^{t-1}}$ rollouts, where $K_t$ is the number of rollouts required at step $t$. In particular, setting $\epsilon = \frac{2}{H+1}$ yields an overall sample complexity of $\mathcal{O}\left(H^3\right)$. In contrast, the baseline target curriculum strategy (with $\alpha_t = \alpha_{\overline{x}}^*$) has an expected overall sample complexity of $\mathcal{O}\left(2^H\right)$.*

*Proof sketch.* Under the target curriculum strategy, the learner receives a non-zero reward only when a successful rollout (reaching the leftmost leaf) occurs. For a uniformly random policy, the probability of following the leftmost branch in a tree of depth $H$ is $p = \left(\frac{1}{2}\right)^H$. Thus, the expected number of rollouts needed to achieve a successful trajectory is $\frac{1}{p} = 2^H$, leading to $\mathcal{O}(2^H)$ sample complexity.

In our curriculum strategy, at each step $t$, the teacher sets the cost threshold $\alpha_t$ so that a fraction $0.5 \cdot (1 - \epsilon)^{t-1}$ of the rollouts yield a reward of 1. For example, at $t = 1$ the teacher selects $\alpha_1$ such that half of the rollouts are successful, ensuring $V^{\pi_1}(\overline{x}; J_{\overline{x}}^{\alpha_1}) \geq 0.5$. As the learner collects $K_t$ rollouts at step $t$, Hoeffding's inequality guarantees that the updated policy $\pi_{t+1}$ satisfies the following: $\mathbb{P}\left[\left|\pi_{t+1}(\text{LEFT} \mid y_0^{(t-1)}) - \pi^*(\text{LEFT} \mid y_0^{(t-1)})\right| > \epsilon\right] \leq 2 \cdot \exp(-2 \cdot K_t \cdot \epsilon^2)$. Setting this probability to be at most $\delta$ and accounting for the compounded $\epsilon$-suboptimality from previous steps,

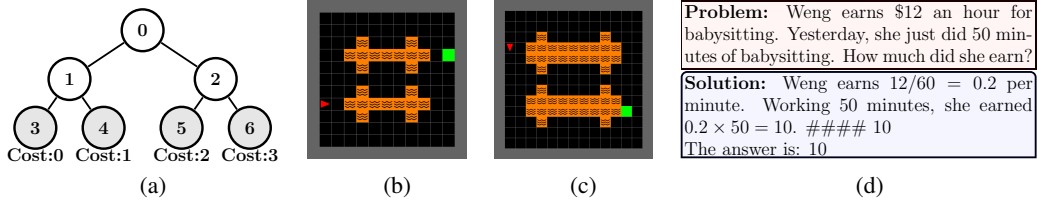

Figure 2: Illustrative visualization of each environment (from left to right): (a) BINARYTREE, (b) PUDDLEGRID-SINGLE, (c) PUDDLEGRID-MULTI, and (d) SVAMP / GSM8K.

one obtains $K_t \geq \frac{\ln\left(\frac{2}{\delta}\right)}{2 \cdot \epsilon^2 \cdot (1-\epsilon)^{t-1}}$. Summing over all $t$ gives the total sample complexity. With a proper choice of $\epsilon$ (e.g., $\epsilon = \frac{2}{H+1}$), one can show that the overall complexity is bounded by $\mathcal{O}(H^3)$. $\qquad\square$

Theorem 1 rigorously shows that our curriculum strategy dramatically reduces the rollout complexity for learning an $\epsilon$-suboptimal policy. In contrast, the baseline suffers from exponential dependence on tree depth $H$, our approach leverages intermediate learning signals to guide the agent more efficiently.

# 4 Experimental Evaluation

We evaluate our curriculum across three RL environments of increasing complexity with challenging trajectory constraints, culminating in a real-world math reasoning task involving an LLM agent. For RL experiments, we employ REINFORCE in both tabular and neural single-/multi-task settings [42]. For LLMs, we adapt Hugging Face's TRL library [43] using the RLOO trainer [44], an RL fine-tuning method equivalent to REINFORCE. This unified setup ensures consistency across all experiments. To assess the robustness of our findings to the choice of the training algorithm, we provide additional results in Appendix E.1, where RL agents are trained with Proximal Policy Optimization (PPO) [45].

## 4.1 Environments

**BINARYTREE.** This environment is inspired by our theoretical analysis (Figure 2a). We use a binary tree of depth $H = 12$, resulting in $2^H = 4096$ leaf nodes, each representing a terminal state associated with a cost. The leftmost leaf has a cost of $0$, with costs increasing monotonically from left to right. The target cost budget is $\alpha^* = 0$, i.e., only the leftmost leaf yields a reward of 1 at test time.

**PUDDLEGRID-SINGLE.** The second environment, PUDDLEGRID-SINGLE (Figure 2b), is a customized variant of the MINIGRID environment. The state representation includes the red agent's location and orientation, while the action space comprises three discrete actions: *move*, *turn-left*, and *turn-right*. The agent's objective is to reach the green square. Stepping on a lava (orange) square increases the trajectory cost by $1$ per time step. The agent receives a reward of 1 only if it reaches the green square with a trajectory cost below the target budget; otherwise, the reward is 0. The target cost budget is set to $\alpha^* = 0$, i.e., the agent must avoid stepping on lava squares entirely.

**PUDDLEGRID-MULTI.** Building on the previous environment, we design PUDDLEGRID-MULTI (Figure 2c), which extends PUDDLEGRID-SINGLE to a multi-task setting. The core structure remains the same, but after each episode, both the agent's initial location and the goal location are randomly selected. The state representation is augmented to include the goal location within the grid. In this multi-task setup, each task can have its own constraint during training and testing. As in PUDDLEGRID-SINGLE, the target cost budget is set to $\alpha^* = 0$.

**SVAMP / GSM8K.** Last, we consider a more challenging setting which involves using LLMs to solve math problems. Given a math problem (Figure 2d), an LLM agent generates a solution trajectory. A reward of 1 is assigned only if the final answer is correct and the number of generated tokens does not exceed the target budget. To ensure the generality of our evaluation, we assess the performance of our algorithm on two mathematical reasoning benchmarks: SVAMP [46], and GSM8K [47]. Given the varying levels of difficulty across datasets, we set the target cost per test sample as a percentage of the base model's original response length. We fine-tune two base models with LoRA [48] using RLOO: QWEN2.5-MATH-1.5B (QWEN) [49], and METAMATH-LLEMMA-7B (METAMATH) [50].

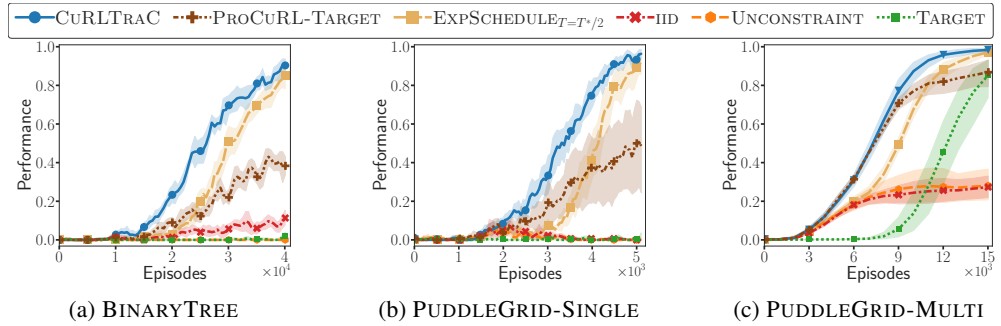

Figure 3: Performance of RL agents trained with different strategies, measured by the agent's mean return (with 95% confidence intervals over 10 random runs), evaluated under test-time constraints.

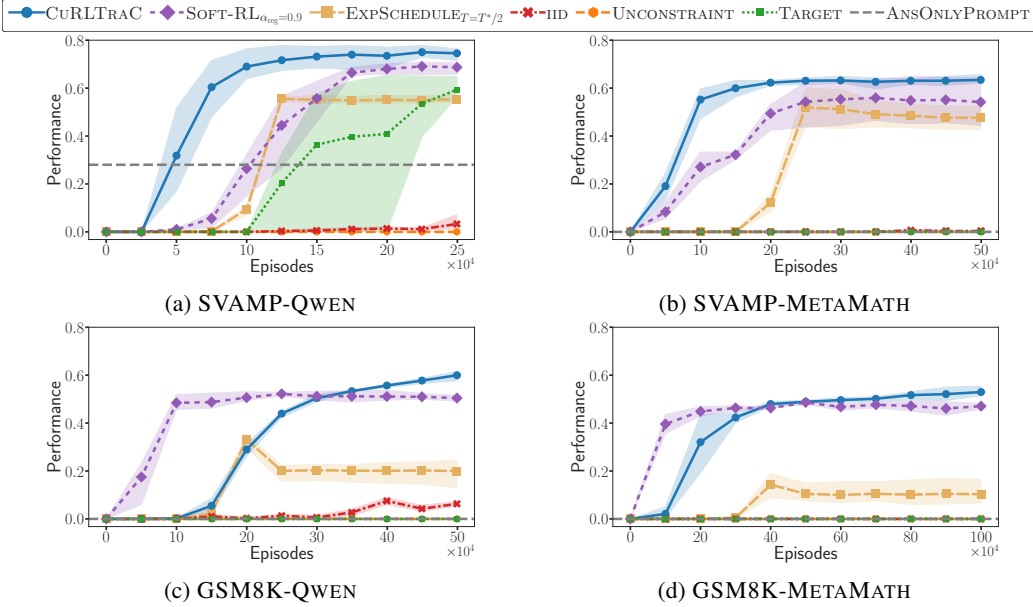

Figure 4: Performance of LLM agents trained with different strategies, measured by the agent's mean accuracy (with 95% confidence intervals over 3 random runs), evaluated under test-time constraints.

## 4.2 Methods Evaluated

**Our strategy CURLTRAC.** We follow Algorithm 2 with a fixed performance threshold of $\beta = 0.5$ across all experiments, following the idea of intermediate difficulty. In Appendix E.2, we conduct a sensitivity analysis of $\beta$ across all RL environments to assess the robustness of the proposed method. Starting from an upper bound on $\alpha$, we perform a binary search to find the smallest value satisfying the constraint in Line 4 of Algorithm 2. A history of per-task rollouts is maintained in a rolling buffer. When task $x$ is selected, its rollout outcome and length are added to the corresponding per-task buffer, and the associated training budget $\alpha_x$ is updated. In practice, updating $\alpha_x$ does not require collecting additional rollouts at different constraint levels. Instead, the update relies solely on previously collected training rollouts, introducing no additional computational overhead.

**Contextual RL curriculum.** We employ a recent curriculum strategy, PROCURL-TARGET [14], which trains RL agents toward challenging target distributions. To apply a contextual RL curriculum in our setting, the trajectory constraint must be treated as an input context, and the deployment-time constraint as the target distribution. While this adaptation is conceptually straightforward, applying it to domains with large context spaces, as in LLMs, is impractical. Effective training with such a technique would require generating all possible (prompt, budget) pairs. Even with discretization, the resulting augmented context space becomes intractable. Therefore, we apply PROCURL-TARGET only in the RL environments.

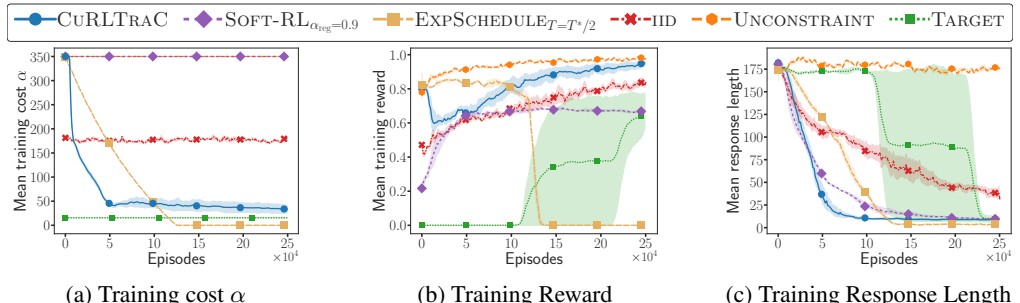

Figure 5: Training plots of SVAMP-QWEN for different strategies. (a) shows the progression of the cost budget $\alpha$ during training. (b) shows the average observed reward during training. (c) shows the progression of the generated response lengths during training.

**Regularized RL baselines.** An alternative option is to relax the strict trajectory-level constrained RL objective in Eq. (1) by considering a regularized, softer version, denoted as SOFT-RL, following [6]. This method aims to balance the trade-off between accuracy and response length rather than explicitly enforcing hard constraints. It can also be applied to our LLM setup. Although it does not guarantee satisfaction of the target constraints at the end of training, the added regularization term encourages both correctness and brevity in the model's responses. While [6] used a coefficient of $\alpha_{\text{reg}} = 0.4$, we found stronger regularization to be more competitive. Hence, we report results with $\alpha_{\text{reg}} = 0.9$, corresponding to SOFT-RL$_{\alpha_{\text{reg}}=0.9}$, to further encourage shorter responses.

**Typical and heuristic baselines.** We compare against three standard baselines: TARGET, trains directly under the deployment constraint; UNCONSTRAINT, trains without restrictions; and IID, trains by sampling the cost budget randomly per episode. Motivated by the observation that CURL-TRAC exhibits an empirical decay pattern resembling exponential scheduling, we also include EXPSCHEDULE [51], a static curriculum with exponential decay. While the decay horizon $T$ is typically unknown in advance, we estimate it based on the convergence behavior of CURLTRAC in each environment. In Appendix E.3, we evaluate EXPSCHEDULE with variable and fixed decay lengths $T = \{T^*, T^*/2, 50000, 5000\}$, where $T^*$ denotes the total number of training episodes per environment. We report results for the most competitive variant, EXPSCHEDULE$_{T=T^*/2}$.

**Prompting-based baseline.** We also consider a prompting-based baseline, ANSONLYPROMPT, to evaluate model's accuracy under test-time constraints without any fine-tuning. ANSONLYPROMPT prompts the model to provide a direct answer without any chain-of-thought reasoning.[2] Further implementation details are provided in Appendix D.2.

### 4.3 Results

**Convergence behavior.** RL and LLM agents trained with CURLTRAC consistently outperform all baselines under test-time constraints (Figures 3 and 4). PROCURL-TARGET performs relatively well in RL environments but relies on providing the target budget as context, limiting its applicability in complex LLM domains. EXPSCHEDULE$_{T=T^*/2}$ effectively trains RL agents in settings where the target budget is identical across tasks; however, its performance becomes inconsistent in per-task target settings, underscoring the need for per-task exponential scheduling. SOFT-RL$_{\alpha_{\text{reg}}=0.9}$, a state-of-the-art method for balancing response length, achieves competitive performance, yet its formulation lacks an explicit notion of target constraints and therefore requires per-domain tuning of its regularization coefficient. ANSONLYPROMPT method shows that, without fine-tuning, the LLM fails to satisfy target constraints.

**Training cost.** Figure 5a shows the progression of the average training cost budget $\alpha$, which reflects task difficulty and defines the training curriculum. SOFT-RL$_{\alpha_{\text{reg}}=0.9}$ incorporates response length directly into the reward function; hence, it does not control $\alpha$. TARGET maintains a fixed cost budget, while UNCONSTRAINT allows maximum token generation. IID samples $\alpha$ randomly per episode. EXPSCHEDULE$_{T=T^*/2}$ decays $\alpha$ at a fixed rate, whereas CURLTRAC adaptively tightens constraints by decreasing $\alpha$ based on model performance, gradually converging to the test target.

---

[2]Note that all other methods use a default chain-of-thought prompt that encourages step-by-step reasoning.

Table 1: Comparison of models in terms of inference metrics when deployed on various consumer hardware configurations. Results are reported for inference on SVAMP test-set with QWEN as base model used for fine-tuning. We consider the following inference metrics: (a) "Response Time", (b) "Response Length", (c) "Accuracy", and (d) "Constr. Accuracy". Here, "Accuracy" denotes the overall test-set accuracy, while "Constr. Accuracy" corresponds to the performance metric used in Figure 4 (i.e., mean accuracy under test-time constraints). We report results for three configurations, namely, M1 (Apple M1 Pro), GTX (Nvidia GTX 1070), and RTX (Nvidia RTX 3060).

| | Inference Metrics | | | | | | | | | | | |
| Method | Response Time (s) | | | Response Length (tokens) | | | Accuracy (%) | | | Constr. Accuracy (%) | | |
| | M1 | GTX | RTX | M1 | GTX | RTX | M1 | GTX | RTX | M1 | GTX | RTX |
|---|---|---|---|---|---|---|---|---|---|---|---|---|
| Base model | 7.5 (0.2) | 3.9 (0.1) | 2.5 (0.1) | 138 (3.9) | 142 (4.0) | 145 (4.2) | 77 | 77 | 75 | 0 | 0 | 0 |
| ANSONLYPROMPT | 5.6 (0.3) | 2.8 (0.1) | 1.7 (0.1) | 104 (5.4) | 102 (5.3) | 104 (5.5) | 60 | 63 | 64 | 9 | 8 | 9 |
| UNCONSTRAINT | 8.1 (0.2) | 4.2 (0.1) | 2.5 (0.1) | 121 (2.1) | 119 (2.2) | 118 (2.1) | 91 | 90 | 91 | 0 | 0 | 0 |
| TARGET | 0.6 (0.1) | 0.5 (0.0) | 0.3 (0.0) | 5 (0.8) | 4 (0.1) | 4 (0.6) | 61 | 61 | 62 | 61 | 61 | 62 |
| CURLTRAC | 0.6 (0.0) | 0.5 (0.0) | 0.3 (0.0) | 3 (0.1) | 3 (0.1) | 3 (0.1) | 74 | 74 | 72 | 74 | 74 | 72 |

**Training reward.** Figure 5b shows that although UNCONSTRAINT and IID achieve increasing training rewards, these gains do not transfer to test-time performance (see Figure 4). TARGET yields delayed or no rewards due to its strict constraints, providing weak learning signals. In contrast, CURLTRAC exhibits an initial reward drop as $\alpha$ decreases and tasks become more challenging, yet the reward remains above the threshold $\beta = 0.5$. Once stabilized around $\beta$, performance improves, indicating that the models learn to adapt to progressively tighter constraints. EXPSCHEDULE$_{T=T^*/2}$ shows a sharp reward decline, indicating rapid constraint tightening. Finally, SOFT-RL$_{\alpha_{reg}=0.9}$ regularizes UNCONSTRAINT, balancing accuracy and brevity through a regularized reward signal.

**Training Response length.** Figure 5c shows varying rates of response-length reduction as models adapt to meet test-time constraints. CURLTRAC achieves the fastest adaptation to shorter responses, followed by SOFT-RL$_{\alpha_{reg}=0.9}$. IID shows a moderate reduction but remains above the target lengths. UNCONSTRAINT and TARGET exhibit no and late change during training.

**LLM Deployment on Consumer Hardware.** Next, we assess the practical impact of our strategy by comparing inference metrics of QWEN models across hardware configurations (Table 1). Performance of the base model serves as a default baseline of the model used for fine-tuning. ANSONLYPROMPT fails to adequately reduce the response time, and displays a drop in accuracy. Fine-tuning with UNCONSTRAINT improves accuracy, but results in high response time; fine-tuning with TARGET decreases response time, but leads to a notable drop in accuracy. Our strategy CURLTRAC achieves a balanced trade-off between response time and accuracy, while also ensuring high constrained accuracy. Relative to the base model, CURLTRAC provides a substantial reduction in response time (e.g., 7.5 vs. 0.6 on M1 configuration), while maintaining accuracy (e.g., 77 vs. 74 on M1 configuration).

## 5 Concluding Discussions

We proposed a curriculum strategy for RL under strict trajectory-level constraints, enabling agents to adapt to stringent resource or safety requirements. Our theoretical analysis on a binary-tree MDP showed faster convergence compared to imposing constraints from the outset, and our empirical results across multiple domains highlighted its effectiveness. Applied to math reasoning, our strategy allowed us to fine-tune an LLM to compress its output chain-of-thought tokens to strict deployment constraints, achieving a substantial speedup during inference time.

Next, we discuss a few limitations and outline a plan for addressing them in future work. First, we assume that deployment-time constraints are predefined. Future work could explore Matryoshka-style learning [52] to train models capable of dynamically adapting to varying target constraints, thereby accommodating different test-time resources. Second, in our setting we compressed the output chain-of-thought tokens of the model to satisfy certain deployment constraints. In future work, it would be interesting to apply this idea of token compression to mitigate other inference-time bottlenecks, such as reducing the number of rollouts in best-of-N sampling or reducing the size of input context. Finally, while our strategy improves performance under constraints, its impact on the model's internal behavior remains unclear. Investigating changes in attention patterns or internal representations after fine-tuning could provide deeper insight into the model's adaptation.

## Acknowledgments and Disclosure of Funding

Funded/Co-funded by the European Union (ERC, TOPS, 101039090). Views and opinions expressed are however those of the author(s) only and do not necessarily reflect those of the European Union or the European Research Council. Neither the European Union nor the granting authority can be held responsible for them.

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

# A    Table of Contents

In this section, we briefly outline the content included in the paper's appendices:

# B    Additional Examples with Model Responses

We present additional qualitative outputs generated by the language model at various stages of training. These examples demonstrate how the fine-tuned model learns to produce compressed responses that explicitly skip steps in chain-of-thought reasoning without compromising final correctness. To maintain clarity and conciseness, we omit intermediate evaluation steps where the model's responses show negligible variation for a given test question. Figure 6 illustrates the progression of METAMATH model responses for a representative example from the SVAMP dataset. In Figure 7, we highlight a notable case where the METAMATH model, by the end of training, produces the correct answer directly, without relying on intermediate chain-of-thought reasoning. Figure 8 shows the progression of QWEN's responses for an example from the GSM8K dataset. The model implicitly halves the total number of glasses and computes the discounted glass price, significantly compressing the chain-of-thought process by reducing the number of generated tokens. Similarly, Figure 9 presents the progression of QWEN's responses for another example from the GSM8K dataset. In this case, the model implicitly computes the number of downloads for the second and third months, and directly proceeds to calculate the total.

---

**Test-set Math Question**: Below is an instruction that describes a task. Write a response that appropriately completes the request. ### Instruction: Jesse's rooms are 19 feet long and 18 feet wide. If she has 20 rooms, how much carpet does she need to cover the floors of all rooms? ### Response: Let's think step by step.

---

**Base Model Response**: Jesse's rooms are 19 feet long and 18 feet wide. To find the area of one room, we multiply the length by the width: 19 feet * 18 feet = 342 square feet. Jesse has 20 rooms in total. To find the total area of all the rooms, we multiply the area of one room by the number of rooms: 342 square feet * 20 = 6840 square feet. Therefore, Jesse needs 6840 square feet of carpet to cover the floors of all the rooms. #### 6840
The answer is: 6840

---

**Fine-tuned Model Response (25K Episodes)**: The area of each room is 19 feet x 18 feet = 342 square feet.Since Jesse has 20 rooms, the total area of all the rooms is 342 square feet x 20 = 6840 square feet.Therefore, Jesse needs 6840 square feet of carpet to cover the floors of all the rooms. #### 6840
The answer is: 6840

---

**Fine-tuned Model Response (50K Episodes)**: The area of each room is 19 feet x 18 feet = 342 square feet.Since Jesse has 20 rooms, she needs a total of 20 x 342 square feet of carpet. #### 6840
The answer is: 6840

---

**Fine-tuned Model Response (75K Episodes)**: She needs 20 * (19 * 18) = 6840 square feet of carpet. #### 6840
The answer is: 6840

---

**Fine-tuned Model Response (100K Episodes)** 19*18*20 #### 6840
The answer is: 6840

---

**Fine-tuned Model Response (250K Episodes)**: 19*18*20
The answer is: 6840

---

Figure 6: METAMATH responses to a SVAMP test question at different stages of training. The model trained with our curriculum technique learns to minimize the chain-of-thought process. In this example, only the final calculation is generated.

**Test-set Math Question**: Below is an instruction that describes a task. Write a response that appropriately completes the request. ### Instruction: Melissa scored a total of 21 points playing some games. Is she scored 7 points in each game. How many games did she play? ### Response: Let's think step by step.

**Base Model Response**: If Melissa scored 7 points in each game, then the total number of games she played is 21 / 7 = 3. Therefore, Melissa played 3 games. #### 3 The answer is: 3

**Fine-tuned Model Response (50K Episodes)**: If Melissa scored 7 points in each game, then the number of games she played is 21 / 7 = 3. #### 3
The answer is: 3

**Fine-tuned Model Response (75K Episodes)**: She played 21/7 = 3 games. #### 3
The answer is: 3

**Fine-tuned Model Response (100K Episodes)**: 21/7 #### 3
The answer is: 3

**Fine-tuned Model Response (225K Episodes)**: 3
The answer is: 3

Figure 7: METAMATH responses to a SVAMP test question at different training stages. By the end of training, the fine-tuned model directly answers without any chain-of-thought tokens.

**Test-set Math Question**: Below is an instruction that describes a task. Write a response that appropriately completes the request. ### Instruction: Kylar went to the store to buy glasses for his new apartment. One glass costs $5, but every second glass costs only 60% of the price. Kylar wants to buy 16 glasses. How much does he need to pay for them? ### Response: Let's think step by step.

**Base Model Response**: To determine how much Kylar needs to pay for 16 glasses, we need to consider the pricing structure: one glass costs $5, but every second glass costs only 60% of $5, which is $3. We can break down the cost as follows:
1. Calculate the cost of the first glass.
2. Calculate the cost of the second glass.
3. Repeat this pattern for all 16 glasses.
4. Sum up the costs to get the total amount Kylar needs to pay.
Let's write the Python code to calculate this.
```python
# Define the cost of a glass and the discounted cost
cost_per_glass = 5 discounted_cost_per_glass = cost_per_glass * 0.6
# Total number of glasses
total_glasses = 16
# Calculate the total cost
total_cost = 0 for i in range(total_glasses): if (i + 1) % 2 == 0: total_cost += discounted_cost_per_glass else: total_cost += cost_per_glass
print(total_cost) ```
``` output 64.0 ``` The total amount Kylar needs to pay for 16 glasses is $\boxed{64}$.

**Fine-tuned Model Response (100K Episodes)**: - The cost of the first glass is $5. - The cost of the second glass is $5 * 0.60 = $3. - The cost of the third glass is $5. - The cost of the fourth glass is $5 * 0.60 = $3. - And so on... We can see that the cost alternates between $5 and $3. Since Kylar wants to buy 16 glasses, there will be 8 glasses at $5 and 8 glasses at $3. The total cost is:

$$8 \times 5 + 8 \times 3 = 40 + 24 = 64$$

So, Kylar needs to pay $\boxed{64}$.

**Fine-tuned Model Response (200K Episodes)**: Cost = 8 * 5 + 8 * 5 * 0.6 = 40 + 24 = 64 $\boxed{64}$

**Fine-tuned Model Response (300K Episodes)**: Total = 8*5 + 8*3 = 40+24=64 $\boxed{64}$

**Fine-tuned Model Response (500K Episodes)**: 8*5+8*3=40+24=64 $\boxed{64}$

Figure 8: QWEN's responses to a GSM8K test question at different training stages. The model trained with our curriculum technique learns to reduce the chain-of-thought process, either by simplifying steps or skipping them. In this example, the model implicitly halves the total number of glasses and computes the discounted glass price, without explicitly generating those intermediate steps.

**Test-set Math Question**: Below is an instruction that describes a task. Write a response that appropriately completes the request. ### Instruction: A new program had 60 downloads in the first month. The number of downloads in the second month was three times as many as the downloads in the first month, but then reduced by 30% in the third month. How many downloads did the program have total over the three months?### Response: Let's think step by step.

**Base Model Response**: To determine the total number of downloads over the three months, we need to calculate the number of downloads for each month and then sum them up. Let's break it down:
1. **First Month:** The program had 60 downloads.
2. **Second Month:** The number of downloads in the second month was three times as many as the downloads in the first month. So, we calculate:

$$3 \times 60 = 180$$

Therefore, the program had 180 downloads in the second month.
3. **Third Month:** The number of downloads in the third month was reduced by 30

$$0.30 \times 180 = 54$$

Then, we subtract this from the second month's downloads:

$$180 - 54 = 126$$

Therefore, the program had 126 downloads in the third month.
4. **Total Downloads:** To find the total number of downloads over the three months, we sum the downloads from each month:

$$60 + 180 + 126 = 366$$

So, the total number of downloads over the three months is $\boxed{366}$.

**Fine-tuned Model Response (100K Episodes)**: 1. The number of downloads in the first month is 60.
2. The number of downloads in the second month is three times the first month, so it is $3 \times 60 = 180$.
3. The number of downloads in the third month is reduced by 30% from the second month, so it is $180 - 0.3 \times 180 = 180 - 54 = 126$.
4. The total number of downloads over the three months is $60 + 180 + 126 = 366$.
The total number of downloads over the three months is $\boxed{366}$.

**Fine-tuned Model Response (200K Episodes)**: First month: 60
Second month: 3 * 60 = 180
Third month: 180 * (1 - 0.30) = 126
Total: 60 + 180 + 126 = 366 $\boxed{366}$

**Fine-tuned Model Response (300K Episodes)**: 3*60=180
180-0.3*180=126
60+180+126=366 $\boxed{366}$

**Fine-tuned Model Response (500K Episodes)**: 60+180+126=366 $\boxed{366}$

Figure 9: QWEN's responses to a GSM8K test question at different training stages. The model trained with our curriculum technique learns to reduce the chain-of-thought process, either by simplifying steps or skipping them. In this example, the model implicitly computes the number of downloads for the second and third months, then directly proceeds to calculate the total.

## C  Proof of Theorem 1

*Proof.* We analyze Algorithm 1 in a single-task setting where the environment consistently selects the same task $\overline{x}$ (Line 3).

**Analysis of the target curriculum strategy ($\beta = 0$).**  Under the target curriculum strategy, the teacher always selects $\alpha_t = \alpha_{\overline{x}}^*$ at each time step $t$, ensuring that the learner consistently trains under the target setting. Initially, until a successful rollout (trajectory reaching the leftmost leaf node) is realized using the learner's current policy, no learning signal is provided to the learner due to the zero reward. Consequently, the learner's policy remains unchanged during this phase.

Once a successful rollout is realized, the learner receives a non-zero reward, which serves as a learning signal. In the best-case scenario, the learner immediately identifies the optimal policy $\pi^*$, given that the reward signal aligns perfectly with the target optimal trajectory. Thus, the sample or rollout complexity of this baseline is determined by the time required to achieve a successful trajectory using the initial random policy.

When sampling from a distribution, the expected number of trials needed to achieve the first success is given by the geometric distribution. If the probability of success in a single trial is $p$, the expected number of trials to get the first success is: $\mathbb{E}\left[\text{Number of trials}\right] = \frac{1}{p}$.

For a random policy, the probability of reaching the leftmost leaf node in a trajectory of depth $H$ is $p = \left(\frac{1}{2}\right)^H$. Consequently, the expected number of rollouts required to achieve a successful trajectory is $\frac{1}{p} = 2^H$. Thus, the total number of steps taken by the learner is $t \cdot K \propto \mathcal{O}\left(2^H\right)$. This highlights the exponential dependence of the baseline's complexity on the depth $H$, reflecting the difficulty of achieving success with this baseline.

**Analysis of the curriculum strategy with $\beta_t = 0.5 \cdot (1 - \epsilon)^{t-1}$.**  We consider the curriculum strategy defined in Eq. ( (2)) and set the threshold parameter as $\beta_t = 0.5 \cdot (1-\epsilon)^{t-1}$, which guarantees that at each time step $t$ the teacher selects a cost budget parameter $\alpha_t$ that facilitates progressive learning by the agent. For the binary-tree MDP under consideration, the optimal value is given by $\max_{\pi,\alpha} V^{\pi}(\overline{x}; J_{\overline{x}}^{\alpha}) = \gamma^H$. Thus, provided that $\gamma$ is sufficiently large—specifically, if $\beta_t \leq \gamma^H$ for all $t$, which is ensured when $\gamma \geq 0.5^{\frac{1}{H}}$—we can simplify the curriculum strategy in Eq. (2) to:

$$\alpha_t \leftarrow \underset{\alpha \in \left[0, \frac{1}{1-\gamma}\right]}{\arg\min} \ (\alpha - \alpha_{\overline{x}}^*)^2 \quad \text{subject to} \quad V^{\pi_t}(\overline{x}; J_{\overline{x}}^{\alpha}) \geq \beta_t.$$

At time step $t = 1$, the teacher selects $\alpha_1$ such that:

$$C_{\overline{x}}\left(y_0^{(0)}, \cdots, y_i^{(H)}\right) \leq \alpha_1, \quad \forall i \in \left\{0, \ldots, 2^{H-1} - 1\right\},$$

$$C_{\overline{x}}\left(y_0^{(0)}, \cdots, y_i^{(H)}\right) > \alpha_1, \quad \forall i \in \left\{2^{H-1}, \ldots, 2^H - 1\right\}.$$

This choice guarantees that, in expectation, half of the rollouts $\xi$ produced by policy $\pi_1$ yield a reward $J_{\overline{x}}^{\alpha_1}(\xi) = 1$ and the other half yield $J_{\overline{x}}^{\alpha_1}(\xi) = 0$. Consequently, the expected value satisfies $V^{\pi_1}(\overline{x}; J_{\overline{x}}^{\alpha_1}) \geq \beta_1 = 0.5$. By the end of step $t = 1$, with a sufficiently large number $K_1$ of rollouts, the learner correctly identifies the optimal action at level 0 with high probability. In particular, by using Hoeffding's inequality, the updated policy $\pi_2$ satisfies the following:

$$\mathbb{P}\left[\left|\pi_2\left(\text{LEFT} \mid y_0^{(0)}\right) - \pi^*\left(\text{LEFT} \mid y_0^{(0)}\right)\right| > \epsilon\right] \leq 2 \cdot \exp\left(-2 \cdot K_1 \cdot \epsilon^2\right) \leq \delta.$$

Thus, with probability at least $1 - \delta$, if $K_1 \geq \frac{\ln\left(\frac{2}{\delta}\right)}{2 \cdot \epsilon^2}$, then:

$$\pi_2\left(\text{LEFT} \mid y_0^{(0)}\right) \geq \pi^*\left(\text{LEFT} \mid y_0^{(0)}\right) - \epsilon = 1 - \epsilon.$$

At step $t = 2$, the teacher selects $\alpha_2$ such that:

$$C_{\overline{x}}\left(y_0^{(0)}, \cdots, y_i^{(H)}\right) \leq \alpha_2, \quad \forall i \in \left\{0, \ldots, 2^{H-2} - 1\right\}$$

$$C_{\overline{x}}\left(y_0^{(0)}, \cdots, y_i^{(H)}\right) > \alpha_2, \quad \forall i \in \left\{2^{H-2}, \ldots, 2^H - 1\right\}.$$

This choice ensures that, in expectation, a fraction $0.5 \cdot (1 - \epsilon)$ of the rollouts $\xi$ generated by policy $\pi_2$ receive a reward $J_{\overline{x}}^{\alpha_2}(\xi) = 1$, with the remainder receiving $J_{\overline{x}}^{\alpha_2}(\xi) = 0$. Therefore, the expected value satisfies $V^{\pi_2}(\overline{x}; J_{\overline{x}}^{\alpha_2}) \geq \beta_2 = 0.5 \cdot (1 - \epsilon)$. By the end of step $t = 2$, with a sufficiently large number $K_2$ of rollouts, the learner identifies the optimal action at level 1 with high probability. In particular, by using Hoeffding's inequality, the updated policy $\pi_3$ satisfies the following:

$$\mathbb{P}\left[\left|\pi_3\left(\text{LEFT} \mid y_0^{(1)}\right) - \pi^*\left(\text{LEFT} \mid y_0^{(1)}\right)\right| > \epsilon\right] \leq 2 \cdot \exp\left(-2 \cdot K_2 \cdot \epsilon^2\right) \leq \delta.$$

Thus, with probability at least $1 - \delta$, if $K_2 \geq \frac{\ln\left(\frac{2}{\delta}\right)}{2 \cdot \epsilon^2 \cdot (1-\epsilon)}$ (where the factor $\frac{1}{1-\epsilon}$ compensates for the $\epsilon$-suboptimality accumulated up to level 0), then:

$$\pi_3\left(\text{LEFT} \mid y_0^{(i)}\right) \geq \pi^*\left(\text{LEFT} \mid y_0^{(i)}\right) - \epsilon = 1 - \epsilon, \quad \forall i \in \{0, 1\}.$$

More generally, at step $t = h$, the teacher selects $\alpha_h$ such that:

$$C_{\overline{x}}\left(y_0^{(0)}, \cdots, y_i^{(H)}\right) \leq \alpha_h, \quad \forall i \in \left\{0, \ldots, 2^{H-h} - 1\right\}$$
$$C_{\overline{x}}\left(y_0^{(0)}, \cdots, y_i^{(H)}\right) > \alpha_h, \quad \forall i \in \left\{2^{H-h}, \ldots, 2^H - 1\right\}.$$

This choice guarantees that, in expectation, a fraction $0.5 \cdot (1 - \epsilon)^{h-1}$ of the rollouts produced by policy $\pi_h$ yield a reward $J_{\overline{x}}^{\alpha_h}(\xi) = 1$, with the remaining rollouts yielding $J_{\overline{x}}^{\alpha_h}(\xi) = 0$; hence, $V^{\pi_h}(\overline{x}; J_{\overline{x}}^{\alpha_h}) \geq \beta_h = 0.5 \cdot (1 - \epsilon)^{h-1}$. By the end of step $t = h$, with a sufficiently large number $K_h$ of rollouts, the learner identifies the optimal action at level $h - 1$ with high probability. Consequently, the updated policy $\pi_{h+1}$ satisfies the following:

$$\pi_{h+1}\left(\text{LEFT} \mid y_0^{(i)}\right) \geq 1 - \epsilon, \quad \forall i \in \{0, \ldots, h-1\},$$

with probability at least $1 - \delta$, provided that $K_h \geq \frac{\ln\left(\frac{2}{\delta}\right)}{2 \cdot \epsilon^2 \cdot (1-\epsilon)^{h-1}}$ (where the factor $\frac{1}{(1-\epsilon)^{h-1}}$ compensates for the $\epsilon$-suboptimality accumulated up to level $h - 2$).

This process continues iteratively until $t = H$, at which point the learner recovers a near-optimal policy $\pi_{H+1}$ satisfying:

$$\pi_{H+1}\left(\text{LEFT} \mid y_0^{(i)}\right) \geq 1 - \epsilon, \quad \forall i \in \{0, \ldots, H-1\},$$

with probability at least $1 - \delta$, provided that $K \geq \frac{\ln\left(\frac{2}{\delta}\right)}{2 \cdot \epsilon^2 \cdot (1-\epsilon)^{H-1}}$.

The total sample complexity for learning an $\epsilon$-suboptimal policy with probability at least $1 - \delta$ is therefore:

$$\sum_{t=1}^{H} K_t = \sum_{t=1}^{H} \frac{\ln\left(\frac{2}{\delta}\right)}{2 \cdot \epsilon^2 \cdot (1 - \epsilon)^{t-1}} \leq H \cdot \frac{\ln\left(\frac{2}{\delta}\right)}{2 \cdot \epsilon^2 \cdot (1 - \epsilon)^{H-1}}.$$

In particular, if we choose $\epsilon = \frac{2}{H+1}$, then $\sum_{t=1}^{H} K_t = \mathcal{O}\left(H^3\right)$. $\qquad\square$

# D  Implementation Details

We provide additional implementation details for the experiments described in Section 4.

## D.1  Compute Resources

We conducted the LLM experiments on a SLURM cluster comprising nodes with eight Nvidia H100 GPUs. Using those resources, the longest LLM experiment ran for approximately three days. We conducted the RL experiments on a cluster comprising nodes equipped with Intel Xeon Gold CPUs.

## D.2  Training Details and Hyperparameters

**RL experiments.** We use the REINFORCE algorithm to train an MLP policy [42]. For BINARYTREE, the policy is tabular, while for PUDDLEGRID-SINGLE and PUDDLEGRID-MULTI, we use a neural policy with two hidden layers. Policies are updated using a batch size of five episodes and the Adam optimizer with a learning rate of $3e-4$ across all environments. PUDDLEGRID-SINGLE and PUDDLEGRID-MULTI have a maximum of 200 steps per episode. Since PUDDLEGRID-MULTI is a multi-task environment, we generated 100 tasks by randomly positioning the agent and goal. To ensure the tasks are challenging, the agent and goal are placed on opposite sides of the lava squares.

**LLM experiments.** As METAMATH-LLEMMA-7B serves as one of the base models in our LLM experiments, we adopt the prompt template (Figure 10), introduced by [50]. It corresponds to a default chain-of-thought prompt that encourages step-by-step reasoning. This prompting template is used across all methods during fine-tuning. Additionally, for both SVAMP [46] and GSM8K [47], we filter out prompts exceeding 512 tokens. To conduct the experiments, we use Huggingface's `Accelerate` and `Transformers` libraries. For RL fine-tuning, we employ the RLOO trainer [44] from the `TRL` library, training with DeepSpeed ZeRO Stage 2 [53]. The RLOO trainer is an adapted version of the PPO trainer; since REINFORCE is a special case of PPO, where the number of PPO epochs and the number of mini-batches are set to 1. This formulation is adopted in the `TRL` implementation. All training parameters and details are reported in Figure 11. Remaining hyperparameters are set to default values and kept consistent across all baselines. Target cost budgets for all tasks are defined as a percentage of the base model's original response length. Given the differing characteristics of models and datasets, we set the target cost budgets to 10% of the original response tokens for the QWEN model, and to 20% and 35% for the METAMATH model on SVAMP and GSM8K, respectively. Regarding the LLM deployment comparison, we evaluate inference metrics using Ollama [54] for both base and fine-tuned models across multiple hardware configurations. All models are converted to GGUF format.

---

**Chain-of-Thought Prompt**

```
Below is an instruction that describes a task.  Write a response
that appropriately completes the request.
### Instruction:
<question>
### Response:  Let's think step by step.
```

Figure 10: Instruction template that encourages step-by-step reasoning used to format the input question. The dataset's math question replaces the `<question>` placeholder. This is the default prompt used across all methods during fine-tuning.

---

**Variants of prompting templates.** We experiment with different prompting templates to examine whether it is possible to improve the performance of large language models under test-time constraints without any fine-tuning. We explore the following variants of prompting templates. In Figure 12, we present a prompt that asks the model to provide an answer directly. In Figure 13, we present a prompt that specifically requests a short answer. In Figure 14, we show a prompt that instructs the model to generate an answer within a specified token budget. The method ANSONLYPROMPT corresponds to the prompting template that achieved the highest test performance.

| Parameter | SVAMP | GSM8K |
|---|---|---|
| Fine-tuning Method | PEFT with LoRA | |
| LoRa $\alpha$ | 96 | |
| LoRa $r$ | 96 | |
| LoRa Dropout | 0.1 | |
| Target Modules | all linear | |
| Per-device Train Batch Size | 4 | 2 |
| Per-device Eval Batch Size | 4 | |
| Gradient Accumulation Steps | 8 | |
| Local Rollout Forward Batch Size | 8 | |
| Learning Rate | $3 \times 10^{-6}$ | |
| KL Coefficient | 0.0001 | |
| Max Response Length | 350 | 512 |

Figure 11: Training details and hyperparameters used across all baselines for SVAMP and GSM8K.

ANSONLYPROMPT

```
Below is an instruction that describes a task.  Write a response
that appropriately completes the request.  Provide a response
that contains only the answer.
### Instruction:
<question>
### Response:
```

Figure 12: Instruction template used to format the input questions to induce direct answers without any chain-of-thought reasoning. The dataset's math question replaces the `<question>` placeholder.

SHORTANSPROMPT

```
Below is an instruction that describes a task.  Write a response
that appropriately completes the request.  Provide a response
that is as short as possible.
### Instruction:
<question>
### Response:
```

Figure 13: Instruction template used to format the input questions to encourage short responses. The dataset's math question replaces the `<question>` placeholder.

ANSUNDERCONSTRAINT

```
Below is an instruction that describes a task.  Write a response
that appropriately completes the request.  Provide a response
where the maximum number of tokens is {target_tokens}.
### Instruction:
<question>
### Response:
```

Figure 14: Instruction template used to format the input questions to encourage answers below a maximum number of tokens that correspond to the target constraint. The dataset's math question replaces the `<question>` placeholder.

**Implementation details about our curriculum strategy CURLTRAC (Section 3).** We apply our curriculum training following Algorithm 1, and update the training cost budget $\alpha_t$ using Algorithm 2.

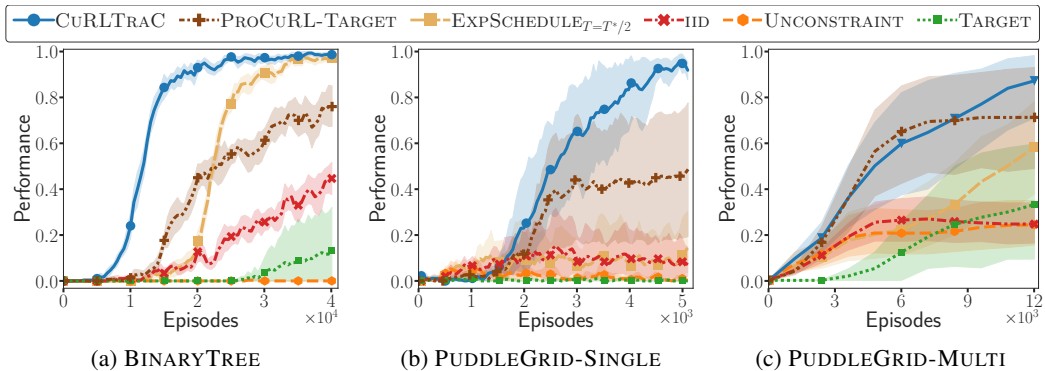

(a) BINARYTREE      (b) PUDDLEGRID-SINGLE      (c) PUDDLEGRID-MULTI

Figure 15: Performance comparison of RL agents trained with PPO as learning algorithm, measured by mean return (with $95\%$ confidence intervals for 10 random runs) under test-time constraints.

As shown, the only hyperparameter is the performance threshold $\beta$. According to Line 3 of Algorithm 2, at each step and for each task $x_t$, the value of $\beta$ is determined by $\min\{\beta, V^{\pi_t}(x_t; J_{x_t}^{\frac{1}{1-\gamma}})\}$, where $V^{\pi_t}(x_t; J_{x_t}^{\frac{1}{1-\gamma}})$ denotes the maximum achievable performance for task $x_t$. We set $\beta = 0.5$ across all environments without any tuning. To avoid performing $K$ rollouts per task in Algorithm 1, we use a buffer to collect rollouts during online training. Several design choices are possible for the buffer. Our strategy, CURLTRAC, employs a fixed-size buffer that stores $K$ rollouts per task. Once the buffer for a given task is full, we update the corresponding $\alpha$ value using a binary search procedure. The buffer then continues to store new rollouts in a FIFO manner, and the $\alpha$ value is recomputed accordingly. The binary search is conducted over a predefined range: the upper bound is set to the maximum trajectory or response length observed in the environment, and the lower bound is defined as the per-task target cost. In our experiments, we use $K = 10$ for all RL and LLM settings, except for SVAMP-QWEN, where $K = 15$ is used.

# E    Additional Experimental Evaluation

## E.1    PPO Experiments

We selected REINFORCE [42] as the primary learning algorithm for our main experiments because it closely aligns with the theoretical analysis. Moreover, REINFORCE-style algorithms have recently gained traction in fine-tuning LLMs, such as in RLOO [44], REINFORCE++, and GRPO, due to their simplicity, efficiency, and stability. Nevertheless, we conducted additional experiments using PPO [45] across all RL environments, and report the results in Figure 15. We observe similar results with REINFORCE, indicating robustness of our technique to the choice of learning algorithm.

## E.2    Parameter Sensitivity Analysis

The performance threshold $\beta$ is the primary hyperparameter of our strategy, CURLTRAC. As mentioned earlier, we use a standard value of $\beta = 0.5$, which aligns with the idea of intermediate difficulty, and keep it fixed across all settings. In general, smaller values of $\beta$ allow for a faster reduction in the training budget $\alpha$, leading to quicker convergence to the target constraint. Conversely, larger values of $\beta$ slow down this progression.

To evaluate the robustness of CURLTRAC, we conduct a sensitivity analysis across all RL environments. In Figure 16, we report performance under test constraints for CURLTRAC with the parameter $\beta$ ranging from 0.1 to 0.9, where $\beta = 0.5$ corresponds to the default setting used in all experiments. Note that $\beta = 0$ and $\beta = 1$ correspond to the TARGET and UNCONSTRAINT baselines, respectively, and are therefore excluded. Figure 16a shows that CURLTRAC is highly robust, performing well across a wide range of $\beta$ values. As expected, increasing $\beta$ slows down the change in training cost budget $\alpha$ (Figure 16b), which in turn affects performance. In Figure 16c, we observe how CURL-TRAC controls the observed reward during training, maintaining it below the respective threshold $\beta$. Additionally, we conduct sensitivity analysis across all RL environments. In Figure 17, we present the

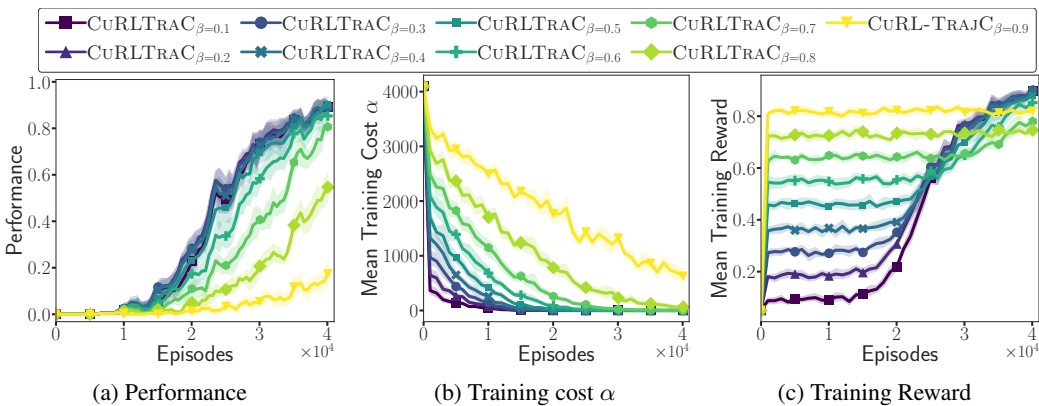

(a) Performance        (b) Training cost $\alpha$        (c) Training Reward

Figure 16: Sensitivity analysis of performance threshold parameter $\beta$ for CURLTRAC strategy in BINARYTREE environment. (a) shows the performance measured by mean return under test-time constraints. (b) shows the progression of the cost $\alpha$ during training. (c) shows the average observed reward during training.

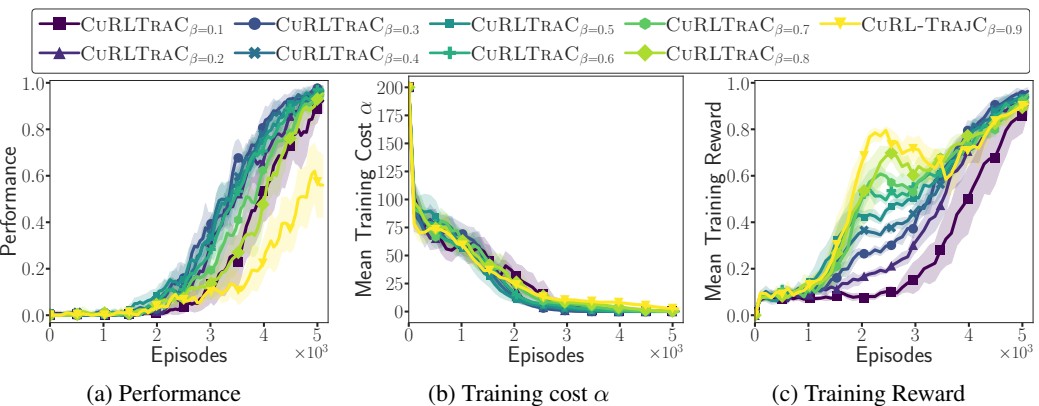

(a) Performance        (b) Training cost $\alpha$        (c) Training Reward

Figure 17: Sensitivity analysis of performance threshold parameter $\beta$ for CURLTRAC strategy in PUDDLEGRID-SINGLE environment. (a) shows the performance measured by mean return under test-time constraints. (b) shows the progression of the cost $\alpha$ during training. (c) shows the average observed reward during training.

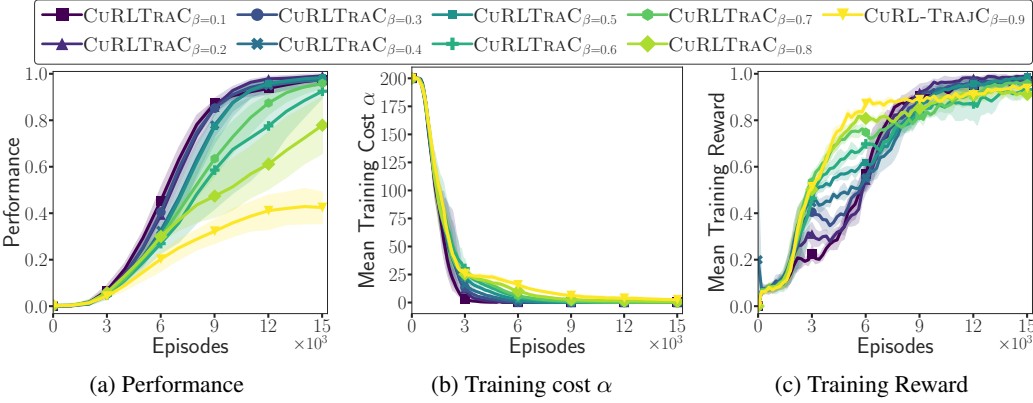

(a) Performance        (b) Training cost $\alpha$        (c) Training Reward

Figure 18: Sensitivity analysis of performance threshold parameter $\beta$ for CURLTRAC strategy in PUDDLEGRID-MULTI environment. (a) shows the performance measured by mean return under test-time constraints. (b) shows the progression of the cost $\alpha$ during training. (c) shows the average observed reward during training.

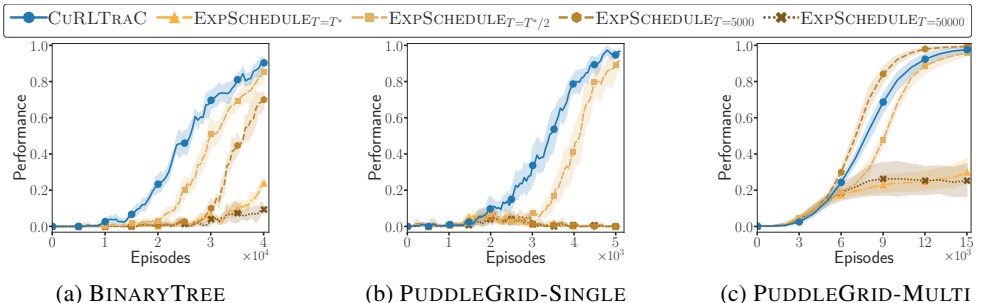

Figure 19: Performance comparison of agents trained with variants of EXPSCHEDULE, measured by mean return (with 95% confidence intervals for 10 random runs) under test-time constraints.

results for the PUDDLEGRID-SINGLE environment, where the robustness of CURLTRAC is evident (Figure 17a). As expected, increasing $\beta$ slows convergence to the test-time target constraints. The training cost trends are similar to those observed in BINARYTREE, though the differences are less pronounced due to the increased complexity of PUDDLEGRID-SINGLE (Figure 17b). The choice of $\beta$ directly influences the level of training reward observed (Figure 17c). Similarly, for multi-task environments, CURLTRAC behaves as expected across different $\beta$ values (Figure 18).

### E.3 Additional Baselines

To provide a more thorough evaluation of our curriculum, we incorporate several additional baselines.

**Variants of EXPSCHEDULE.** As illustrated in Figure 5, our curriculum strategy naturally produces per-task $\alpha$ schedules that, on average, follow a form of exponential decay. This observation motivates the inclusion of a related baseline, EXPSCHEDULE, which applies a fixed exponential decay schedule [51]. The intuition is that a well-tuned exponential decay should perform comparably to our strategy. We evaluate variants of this strategy with different decay lengths $T = \{T^*, T^*/2, 50000, 5000\}$, where $T^*$ denotes the total number of training episodes per environment. These correspond to EXPSCHEDULE$_{T=T^*}$, EXPSCHEDULE$_{T=T^*/2}$, EXPSCHEDULE$_{T=50000}$, and EXPSCHEDULE$_{T=5000}$, respectively. Figure 19 compares our strategy, CURLTRAC, with the variants of EXPSCHEDULE across RL environments. Our approach consistently achieves the strongest and most stable empirical performance. In contrast, the fixed exponential schedules exhibit instability, yielding competitive results in some environments but failing to generalize across all. Among the exponential variants, EXPSCHEDULE$_{T=T^*/2}$ demonstrates comparatively greater stability, albeit with lower overall performance than CURLTRAC. This suggests that incorporating domain knowledge, such as the total number of training episodes $T^*$, plays a critical role in tuning the decay schedule effectively.

**CURLTRAC-GLOBAL.** This is a variant of CURLTRAC that maintains a single global value of $\alpha$ shared across all tasks, rather than assigning an individual training budget $\alpha_x$ per task $x$. We refer to this method as CURLTRAC-GLOBAL. The procedure is identical to CURLTRAC, except that rollouts from all selected tasks are stored in a common global buffer. As a result, a single global training budget $\alpha$ is updated after each task. In single-task settings (as in BINARYTREE and PUDDLEGRID-SINGLE), the two strategies are equivalent. The adaptive CURLTRAC requires collecting a sufficient number of rollouts per task to enable task-specific updates of $\alpha$. In contrast, one advantage of CURLTRAC-GLOBAL is a shorter warm-up phase, as the global buffer is populated with rollouts from multiple tasks, allowing the threshold $\alpha$ to begin adapting earlier. We provide a direct comparison between CURLTRAC and its global variant, CURLTRAC-GLOBAL.

**Performance comparison with additional baselines.** In Figure 20, we compare the convergence behavior of LLM agents across additional baselines, including the global strategy CURLTRAC-GLOBAL, a softer regularization variant SOFT-RL$_{\alpha_{\text{reg}}=0.4}$, and a fixed decay rate strategy EXPSCHEDULE$_{T=50000}$. Table 2 summarizes test-time performance at two training checkpoints. The strategy CURLTRAC-GLOBAL shows a rapid initial performance increase, as it does not require a warm-up phase for collecting rollouts per task. However, due to per-task targets in the LLM setting, its performance ultimately plateaus below that of the adaptive CURLTRAC. The weaker regularization in SOFT-RL$_{\alpha_{\text{reg}}=0.4}$ leads to slower convergence toward test-time constraints compared to SOFT-RL$_{\alpha_{\text{reg}}=0.9}$, as expected. The EXPSCHEDULE$_{T=50000}$ strategy is unstable across all settings.

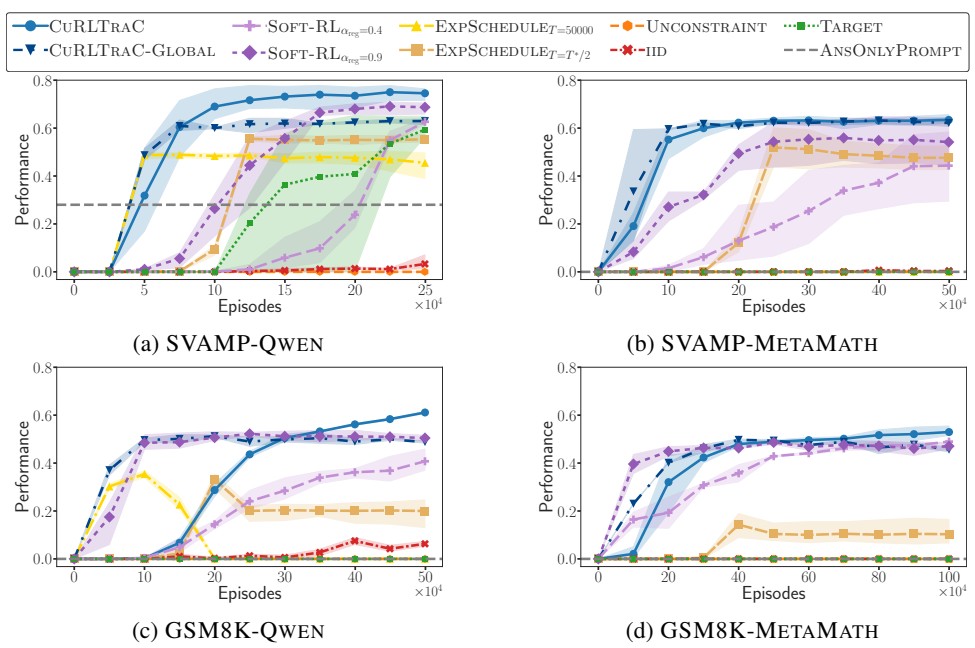

Figure 20: Performance of LLM agents trained with different strategies, measured by the agent's mean accuracy (with 95% confidence intervals over 3 random runs), evaluated under test-time constraints.

Table 2: Summary of LLM performance across two mathematical benchmarks, SVAMP and GSM8K, and two large language models, QWEN2.5-MATH-1.5B and METAMATH-LLEMMA-7B, as detailed in Figure 20. For each column, the highest-performing strategy is shown in bold, and the second-best is underlined.

| Method | Dataset: SVAMP | | | | Dataset: GSM8K | | | |
| --- | --- | --- | --- | --- | --- | --- | --- | --- |
| | Model: QWEN | | Model: METAMATH | | Model: QWEN | | Model: METAMATH | |
| | # Episodes | | # Episodes | | # Episodes | | # Episodes | |
| | 100K | 250K | 250K | 500K | 250K | 500K | 500K | 1M |
| | **Test-time Performance** | | | | | | | |
| CURLTRAC | **0.69** | **0.74** | **0.63** | **0.63** | 0.43 | **0.60** | **0.49** | **0.53** |
| CURLTRAC-GLOBAL | 0.60 | 0.63 | 0.62 | 0.62 | 0.49 | 0.48 | **0.49** | 0.46 |
| SOFT-RL$_{\alpha_{reg}=0.4}$ | 0.00 | 0.63 | 0.19 | 0.44 | 0.24 | 0.41 | 0.43 | 0.49 |
| SOFT-RL$_{\alpha_{reg}=0.9}$ | 0.26 | 0.69 | 0.54 | 0.54 | **0.52** | 0.50 | **0.49** | 0.47 |
| EXPSCHEDULE$_{T=50000}$ | 0.48 | 0.45 | 0.00 | 0.00 | 0.00 | 0.00 | 0.00 | 0.00 |
| EXPSCHEDULE$_{T=T^*/2}$ | 0.10 | 0.55 | 0.52 | 0.48 | 0.20 | 0.20 | 0.10 | 0.10 |
| IID | 0.00 | 0.03 | 0.00 | 0.00 | 0.01 | 0.06 | 0.00 | 0.00 |
| UNCONSTRAINT | 0.00 | 0.00 | 0.00 | 0.00 | 0.00 | 0.00 | 0.00 | 0.00 |
| TARGET | 0.00 | 0.59 | 0.00 | 0.00 | 0.00 | 0.00 | 0.00 | 0.00 |
| ANSONLYPROMPT | 0.28 | 0.28 | 0.00 | 0.00 | 0.00 | 0.00 | 0.00 | 0.00 |

