# OpenReview forum: "Curriculum Design for Trajectory-Constrained Agent: Compressing Chain-of-Thought Tokens in LLMs"
_NeurIPS.cc/2025/Conference — NeurIPS 2025 poster_

### Official Review · Reviewer_UKTS · 2025-06-17

**Clarity:** 3
**Significance:** 2
**Originality:** 4
**Rating:** 4
**Confidence:** 4

**Summary:**

The paper suggests a curriculum learning strategy to learn the textual reasoning chain-of-thoughts limited to a certain target length (in tokens). The proposed CURL-TRAJC strategy is based on the properties of a binary tree and is given theoretical justification. The supporting experiments are performed on 3 synthetic/toy benchmarks and 2 math benchmarks. The paper aims to improve LLM’s end-user experience with LLM inference in exchange for extra complexity in the training stage.

**Questions:**

[1] What is “deployment”? This term from the industry/business is not used in academia often.

[2] Is there any way to preserve the reward even with squeezing the size of reasoning into the constraint?

[3] How could tool invocations be incorporated into reasoning?

**Ethical Concerns:**

["NO or VERY MINOR ethics concerns only"]

**Final Justification:**

The authors have addressed my concerns.

**Limitations:**

The method is shown to work only for math problems. The method is limited to text-only reasoning, without support for within-reasoning tool invocation.

**Quality:**

3

**Strengths And Weaknesses:**

Strengths:

[1] The study is well-rounded, with a theoretically clear problem statement, rigorous theoretical analysis of the solution and the support by experiments.

Weaknesses:

[1] The background for the problem is too brief. While the example on Figure 1 is very insightful, it is not clear, what could other practical uses be apart from solving math puzzles.

[2] The experiments are performed on 3 synthetic/toy benchmarks and the only two practically significant benchmarks (SVAMP and GSM8K) are both math, which is not diverse enough to prove the generalization of the method. A non-math benchmark should be considered (as the authors admit in the discussions), for example HumanEval or even MLEBench.

[3] Phrasing “Compressing LLM Output Tokens” is highly misleading. The individual tokens are not compressed, but rather the reasoning as a sequence of tokens is learned to be shorter.

[4] “Response after 400K Episodes” suggests that training is very slow. 400K episodes looks like a lot to teach the model to reason briefly. Curriculum learning makes a promise to make training fast, which does not seem to be the case here.

[5] Figure 4, CuRL-TrajC clearly is able to get to the target cost the fastest, however its final rewardlags behind the Unconstrained by a large margin, and is not better than other baselines. It is expected for the tradeoff to be like that, but I do not see much of the advantage of CuRL-TrajC compared to ExpSchedule. Yes, it arrives to the same brief reasoning and poor reward, but faster.

---

> ### Author Rebuttal · Authors · 2025-07-30
>
> Thank you for carefully reviewing our paper! We greatly appreciate your feedback. Please see below our responses to your comments.
>
> -----
>
> **1. The background for the problem is too brief. While the example on Figure 1 is very insightful, it is not clear, what could other practical uses be apart from solving math puzzles.**
>
> We would like to note that our problem setup — and the proposed curriculum strategy — applies broadly to scenarios where trajectory-level constraints can be expressed via a cost function over the output or response trajectories. More concretely:
> - Reaching a goal under navigation cost constraints, as demonstrated in the paper through the agent's path cost in MiniGrid.
> - LLM output generation with reduced reasoning tokens, also covered in the paper as trajectory length constraints.
> - In the rebuttal, we present additional LLM experiments involving variable token length constraints based on the original model response length.
>
> Beyond these, our method naturally extends to:
> - Safety constraints in RL domains, modeled via safety cost functions over the agent’s trajectory.
> - Harmfulness constraints in LLMs, where a harmfulness or toxicity score can serve as the cost function.
> - Code generation tasks, where generated programs must satisfy test cases or meet resource/memory constraints — both of which can be captured via cost functions over the generated output.
>
> In general, any constraint on response trajectories that can be formalized as a cost function is compatible with our framework. While it’s not practical to include all possible scenarios in the paper, we aimed to showcase a representative set of impactful applications.
>
> -----
>
> **2. The experiments are performed on 3 synthetic/toy benchmarks and the only two practically significant benchmarks (SVAMP and GSM8K) are both math … A non-math benchmark should be considered (as the authors admit in the discussions), for example HumanEval or even MLEBench.**
>
> As noted above, we conducted experiments with different types of trajectory-level constraints. We focused on mathematical tasks because they are widely regarded as standard benchmarks for evaluating the reasoning capabilities of LLMs. To ensure diversity within this domain, we included two distinct math datasets: SVAMP and GSM8K.
>
> Furthermore, we have conducted additional LLM-based experiments, which will be included in the revised version of the paper.
>
> The new experiments include:
> - A model of different family and size — Qwen2.5-Math-1.5B — evaluated on both datasets.
> - A variable target setting, where each task $t$ has its own target budget $\alpha^*$, determined based on the original model response length. This setting is applied across both models and datasets, resulting in four additional experiments.
>
> We present the results in tabular form below.
>
> **Experiments with Qwen2.5-Math-1.5B: SVAMP**
>
> | Method | 50K | 100K | 150K | 200K | 250K |
> | ----- | --- | --- | --- | --- | --- |
> | CuRL-TrajC (ours) | $0.59$ | $0.78$ | $0.81$ | $0.80$ | $0.81$ |
> | ExpSchedule | $0.00$  | $0.00$ | $0.00$ |  $0.00$ | $0.64$ |
> | IID | $0.00$  | $0.00$ | $0.02$ | $0.20$ | $0.29$ |
> | Unconstrained | $0.00$ | $0.00$ | $0.00$ | $0.00$ | $0.00$ |
> | Target | $0.00$ | $0.00$ | $0.57$ | $0.61$ | $0.60$ |
> |
>
> **Experiments with Qwen2.5-Math-1.5B: GSM8K**
>
> | Method | 200K | 300K | 400K | 500K |
> | ----- | --- | --- | --- | --- |
> | CuRL-TrajC (ours) | $0.37$ | $0.56$ | $0.58$ | $0.62$ |
> | ExpSchedule | $0.00$  | $0.00$ | $0.04$ |  $0.50$ |
> | IID | $0.02$  | $0.03$ | $0.13$ | $0.17$ |
> | Unconstrained | $0.00$ | $0.00$ | $0.00$ | $0.00$ |
> | Target | $0.23$ | $0.25$ | $0.25$ | $0.25$ |
> |
>
> **Experiments with variable targets Qwen2.5-Math-1.5B: SVAMP**
>
> | Method | 50K | 100K | 150K | 200K | 250K |
> | ----- | --- | --- | --- | --- | --- |
> | CuRL-TrajC (ours) | $0.27$ | $0.76$ | $0.77$ | $0.77$ | $0.77$ |
> | ExpSchedule | $0.00$  | $0.00$ | $0.00$ |  $0.00$ | $0.60$ |
> | IID | $0.00$  | $0.00$ | $0.01$ | $0.02$ | $0.07$ |
> | Unconstrained | $0.00$ | $0.00$ | $0.00$ | $0.00$ | $0.00$ |
> | Target | $0.00$ | $0.00$ | $0.00$ | $0.00$ | $0.00$ |
> |
>
> **Experiments with variable targets Qwen2.5-Math-1.5B: GSM8K**
>
> | Method | 200K | 300K | 400K | 500K |
> | ----- | --- | --- | --- | --- |
> | CuRL-TrajC (ours) | $0.26$ | $0.49$ | $0.56$ | $0.61$ |
> | ExpSchedule | $0.00$  | $0.00$ | $0.01$ |  $0.23$ |
> | IID | $0.01$  | $0.02$ | $0.08$ | $0.05$ |
> | Unconstrained | $0.00$ | $0.00$ | $0.00$ | $0.00$ |
> | Target | $0.00$ | $0.00$ | $0.00$ | $0.00$ |
> |
>
> **Experiments with variable targets MetaMath: SVAMP**
>
> | Method | 50K | 100K | 150K | 200K | 250K |
> | ----- | --- | --- | --- | --- | --- |
> | CuRL-TrajC (ours) | $0.06$ | $0.46$ | $0.61$ | $0.63$ | $0.63$ |
> | ExpSchedule | $0.00$  | $0.00$ | $0.00$ |  $0.00$ | $0.58$ |
> | IID | $0.00$  | $0.00$ | $0.00$ | $0.01$ | $0.00$ |
> | Unconstrained | $0.00$ | $0.00$ | $0.00$ | $0.00$ | $0.00$ |
> | Target | $0.00$ | $0.00$ | $0.00$ | $0.00$ | $0.00$ |
> |
>
> **Experiments with variable targets MetaMath: GSM8K**
>
> | Method | 200K | 300K | 400K | 500K |
> | ----- | --- | --- | --- | --- |
> | CuRL-TrajC (ours) | $0.18$ | $0.38$ | $0.41$ | $0.42$ |
> | ExpSchedule | $0.00$  | $0.00$ | $0.00$ |  $0.27$ |
> | IID | $0.00$  | $0.00$ | $0.00$ | $0.00$ |
> | Unconstrained | $0.00$ | $0.00$ | $0.00$ | $0.00$ |
> | Target | $0.00$ | $0.00$ | $0.00$ | $0.00$ |
> |
>
> -----
>
> **3. Phrasing “Compressing LLM Output Tokens” is highly misleading … the reasoning as a sequence of tokens is learned to be shorter.**
>
> In the revised version, we plan to rephrase the title to “Curriculum Design for Trajectory-Constrained Agents with Applications to Efficient Reasoning Compression in LLMs” to make it more explicit that we refer to the reasoning trajectory of the model and not to the individual tokens. We thank the reviewer for the suggestion.
>
> -----
>
> **4. “Response after 400K Episodes” suggests that training is very slow … Curriculum learning makes a promise to make training fast, which does not seem to be the case here.**
>
> In general, curriculum learning aims to improve convergence and enable learning of complex tasks that might otherwise be unlearnable. In our case, the final constrained task is highly challenging, and baseline curriculum strategies fail to learn it successfully. Notably, compared to other methods that do converge — such as ExpSchedule — our approach, CuRL-TrajC, converges significantly faster, demonstrating clear advantages in training efficiency.
>
> -----
>
> **5. Figure 4, CuRL-TrajC clearly is able to get to the target cost the fastest … Yes, it arrives to the same brief reasoning and poor reward, but faster.**
>
> The horizontal line represents an upper bound — showing the model’s performance when trained under an unconstrained objective purely to maximize reward. As such, it is not directly comparable to the constrained setting, where the objective includes both performance and cost.
>
> Importantly, compared to ExpSchedule, CuRL-TrajC converges significantly faster. For context, the performance gap corresponds to approximately 250K episodes — or about 43 hours of training using 4 GPUs. Furthermore, as shown in our additional results presented during rebuttal, CuRL-TrajC consistently outperforms ExpSchedule across all benchmarks.
>
> -----
>
> **6. What is “deployment”? This term from the industry/business is not used in academia often.**
>
> We use test-time and deployment interchangeably throughout the paper.
>
> -----
>
> **7. Is there any way to preserve the reward even with squeezing the size of reasoning into the constraint?**
>
> It is expected that overall accuracy may drop when the response length is heavily constrained. However, our results show that it's possible to achieve substantial output compression — leading to faster inference — while still maintaining competitive performance relative to baselines.
>
> Several strategies can help improve the trade-off:
> - The compression level can be adapted per task, based on task difficulty and the model’s initial performance.
> - Since the fine-tuned model generates shorter responses, techniques like Best-of-N (BoN) sampling can be used at inference time to boost performance without increasing overall cost. For example, if the original model outputs 120 tokens on average and the compressed model outputs 20, using best-of-5 sampling remains more efficient.
> - Further improvements could come from tuning training hyperparameters, such as the KL coefficient or LoRA configuration.
>
> -----
>
> **8. The method is limited to text-only reasoning, without support for within-reasoning tool invocation. How could tool invocations be incorporated into reasoning?**
>
> We would like to clarify that our method is not limited to text-only reasoning — see, for example, the safe navigation experiments in RL. The framework is generally applicable to scenarios where trajectory-level constraints can be expressed as cost functions over output trajectories.
> Tool invocations can be incorporated similarly. If invoking a tool (e.g., a calculator) incurs an additional cost, this can be captured in the total trajectory cost. For simpler tasks, the model may choose to reason without invoking the tool to minimize cost, while for harder tasks, it may opt to bear the additional cost for better performance. This fits naturally within our trajectory-level cost formulation.
>
> -----
>
> We hope that our responses can address your concerns and are helpful in improving your rating. If you have any other comments or feedback, please let us know! We are looking forward to hearing back from you! Thank you again for the review.

---

> > ### Comment · Reviewer_UKTS · 2025-08-06
> >
> > Dear authors, thank you for your rebuttal. I have increased my score.

---

> > > ### Author Response · Authors · 2025-08-08
> > >
> > > We sincerely thank the reviewer for their engagement during the discussion period and appreciate the reviewer's input in helping us improve the paper.

---

### Official Review · Reviewer_jEUn · 2025-07-03

**Clarity:** 3
**Significance:** 3
**Originality:** 3
**Rating:** 5
**Confidence:** 2

**Summary:**

This paper introduces a curriculum learning strategy for training agents to operate deployment time constraints like limited resources or safety requirements. The method adaptively adjusts cost budget during training, starting off with more relaxed constraints and then slowly makes the constraints more restrictive. This helps smoothen the transition into the more strict deployment conditions. The paper provides theoretical analysis that demonstrates the curriculum strategy accelerates training compared to baseline approaches that enforce the strict constraints right from the start. Empirically, the method is tested on reinforcement learning and LLM agents and achieves significant inference time speedup for LLM output token compression hardware.

**Questions:**

Questions
- does the teacher component incur an additional computational overhead?
- is token length =40 a real world constraint? I suppose constraints like memory footprint or harmfulness scores would be considered a more meaningful constraints in the semantic experiments of the LLM space
- how would you choose for a new task what the best maximal/relaxed cost budget to start off with is? Is it always 1/(1-gamma)?

**Ethical Concerns:**

["NO or VERY MINOR ethics concerns only"]

**Final Justification:**

My main concern was with the baselines -- specifically I recommended the authors consider sota RL algorithms to compare such as PPO. The authors have provided these results during the rebuttal.

**Limitations:**

yes

**Quality:**

3

**Strengths And Weaknesses:**

Strengths
- the problem is well-motivated and the writing is generally clear
- while having an adaptive budget for curriculum isn’t exactly a novel concept [1], I think examining the empirical performance of adapting costs in application to RL for LLM is quite interesting
- the approach is promising in that it demonstrates significant performance improvements compared to baseline in highly complex tasks like solving gsm8k reasoning questions
- overall the approach is theoretically grounded in computational/sample complexity

Weakness
- for th rl experiments it would have been nice to see experiments with a bit more complex algorithms than reinforce, like sota ones including ppo
- also safe rl environments that are more complex than puddle-grid would be valuable
- (minor) it would be nice to have a table of contents of all the variables defined (perhaps in appendix) for improved readability
- (minor) in the figures, you should indicate which algorithm is the papers with something like (ours) to distinguish from baselines

[1]Narvekar et al. "Curriculum learning for reinforcement learning domains: A framework and survey." JMLR 2020

---

> ### Author Rebuttal · Authors · 2025-07-30
>
> Thank you for carefully reviewing our paper! We greatly appreciate your feedback. Please see below our responses to your comments.
>
> -----
>
> **1. For the RL experiments it would have been nice to see experiments with a bit more complex algorithms than reinforce, like sota ones including ppo.**
>
> We have conducted additional experiments using PPO across all RL environments and report the results below.
>
> **Experiments with PPO algorithm: BinaryTree**
>
> | Method | 10K | 20K | 30K | 40K |
> | ----- | --- | --- | --- | --- |
> | CuRL-TrajC (ours) | $0.30$ | 0.93 | $0.97$ | $0.99$ |
> | ProCuRL-Target | $0.02$ | $0.50$ | $0.63$ | $0.74$ |
> | ExpSchedule | $0.00$  | $0.04$ | $0.10$ |  $0.54$ |
> | IID | $0.01$  | $0.16$ | $0.26$ | $0.44$
> | Unconstrained | $0.00$ | $0.00$ | $0.00$ | $0.00$ |
> | Target | $0.00$ | $0.00$ | $0.05$ | $0.15$ |
> |
>
> **Experiments with PPO algorithm: PuddleGrid-Single**
>
> | Method | 2K | 3K | 4K | 5K |
> | ----- | --- | --- | --- | --- |
> | CuRL-TrajC (ours) | $0.25$ | $0.60$| $0.78$ | $0.83$ |
> | ProCuRL-Target | $0.18$ | $0.30$ | $0.33$ | $0.47$ |
> | ExpSchedule | $0.10$  | $0.06$ | $0.05$ |  $0.06$ |
> | IID | $0.09$  | $0.07$ | $0.10$ | $0.08$
> | Unconstrained | $0.04$ | $0.02$ | $0.01$ | $0.00$ |
> | Target | $0.01$ | $0.00$ | $0.01$ | $0.00$ |
> |
>
> **Experiments with PPO algorithm: PuddleGrid-Multi**
>
> | Method | 2K | 5K | 8K | 10K |
> | ----- | --- | --- | --- | --- |
> | CuRL-TrajC (ours) | $0.36$ | $0.65$ | $0.84$ | $0.90$ |
> | ProCuRL-Target | $0.41$ | $0.68$ | $0.72$ | $0.71$ |
> | ExpSchedule | $0.31$  | $0.30$ | $0.33$ |  $0.46$ |
> | IID | $0.24$  | $0.27$ | $0.25$ | $0.25$
> | Unconstrained | $0.20$ | $0.20$ | $0.26$ | $0.24$ |
> | Target | $0.00$ | $0.20$ | $0.30$ | $0.36$ |
> |
>
> We selected REINFORCE as the primary learning algorithm for our experiments because it closely aligns with the theoretical analysis presented in the paper. Moreover, REINFORCE-style methods have recently gained traction in fine-tuning LLMs — such as in RLOO [1], REINFORCE++ [2], and GRPO — due to their simplicity, efficiency, and stability. Notably, in our LLM experiments, we use the RLOO training algorithm.
>
> [1] Ahmadian et al., “Back to Basics: Revisiting REINFORCE Style Optimization for Learning from Human Feedback in LLMs”, arXiv preprint arXiv:2402.14740 (2024).
>
> [2] Hu et al., "REINFORCE++: An Efficient RLHF Algorithm with Robustness to Both Prompt and Reward Models.", arXiv preprint arXiv:2501.03262 (2025).
>
> [3] Shao et al., "DeepSeekMath: Pushing the Limits of Mathematical Reasoning in Open Language Models." arXiv preprint arXiv:2402.03300 (2024).
>
> -----
>
> **2. Safe RL environments that are more complex than puddle-grid would be valuable.**
>
> Minigrid-style environments such as PuddleGrid-Single and PuddleGrid-Multi have been used in recent safe reinforcement learning literature ([1], [2]) as standard benchmarks for studying goal-reaching tasks under navigation cost constraints. We consider these environments sufficiently rich to evaluate our proposed method and to provide meaningful insights into its effectiveness. Moreover, our work includes a diverse range of experiments involving both RL and LLM agents. That said, we agree that exploring additional complex environments would be an interesting direction for future work.
>
> [1] Kim et al., “Sample-Efficient and Safe Deep Reinforcement Learning via Reset Deep Ensemble Agents”, NeurIPS 2023.
>
> [2] Gros et al., “Safe Reinforcement Learning Through Regret and State Restorations in Evaluation Stages”, AAAI 2024.
>
> -----
>
> **3. Does the teacher component incur an additional computational overhead?**
>
> The teacher selects the current training budget $\alpha$ using a simple binary search procedure, which incurs minimal computational overhead. Importantly, it relies solely on rollout histories already collected by the student during training, so no additional rollouts are required — making the budget estimation effectively cost-free.
>
> -----
>
> **4. Is token length =40 a real world constraint? I suppose constraints like memory footprint or harmfulness scores would be considered a more meaningful constraints in the semantic experiments of the LLM space.**
>
> A constraint on response length naturally arises in scenarios where users require faster responses or reduced inference cost and time. Fine-tuning LLMs to produce more concise, less verbose reasoning can lead to more efficient and practical deployments. Thus, we view length constraints as a meaningful and realistic use case.
>
> We would like to note that our problem setup — and the proposed curriculum strategy — applies broadly to scenarios where trajectory-level constraints can be expressed via a cost function over the output or response trajectories.
>
> For instance, our method can extend to:
> - Harmfulness constraints in LLMs, where a harmfulness or toxicity score can represent the cost function.
> - Programming tasks, where generated programs must satisfy certain test-cases or meet resource/memory constraints — both of which can be captured via cost functions over the generated output.
>
> In general, any constraint on response trajectories that can be formalized as a cost function is compatible with our framework. While it’s not practical to include all possible scenarios in the paper, we aimed to showcase a representative set of impactful applications.
>
> -----
>
> **5. How would you choose for a new task what the best maximal/relaxed cost budget to start off with is? Is it always 1/(1-gamma)?**
>
> While setting the maximum cost budget to $1/(1-\gamma)$ follows from the theoretical analysis, in practice, there are more intuitive and task-specific ways to define it. One simple option is to choose a sufficiently large value when no prior knowledge is available, ensuring that early learning is unconstrained. Alternatively, the budget can be set based on the maximum expected cost observed in task trajectories. For example, in LLM settings with length constraints, if generation is limited to $N$ tokens with a per-token cost of 1, a natural choice for the maximum budget is $N$. Similarly, in RL domains, the maximum budget can correspond to the episode length or the maximum number of environment steps.
>
> -----
>
> (Minor suggestions)
> **6. It would be nice to have a table of contents of all the variables defined (perhaps in appendix) for improved readability … In the figures, you should indicate which algorithm is the papers with something like (ours) to distinguish from baselines.**
>
> We thank the reviewer for the suggestions. We will incorporate them in the revised version.
>
> -----
>
> We hope that our responses can address your concerns and are helpful in improving your rating. If you have any other comments or feedback, please let us know! We are looking forward to hearing back from you! Thank you again for the review.

---

> > ### Author Response · Authors · 2025-08-06
> >
> > As the discussion period is ending soon, we are writing to thank the reviewer for their constructive feedback. We hope that our responses have addressed your concerns and are helpful in improving your rating. We will incorporate the reviewer's feedback and our responses in the updated paper.

---

> > ### Comment · Reviewer_jEUn · 2025-08-08
> >
> > Thanks to the authors for their detailed rebuttal; I have increased my score

---

> > > ### Author Response · Authors · 2025-08-08
> > >
> > > We sincerely thank the reviewer for their engagement during the discussion period and appreciate the reviewer's input in helping us improve the paper.

---

> ### Comment · Area_Chair_oJ7t · 2025-08-06
>
> Please remember to respond to authors' rebuttal as soon as possible.
>
> Thank you!
>
> -AC

---

### Official Review · Reviewer_CkZL · 2025-07-03

**Clarity:** 3
**Significance:** 3
**Originality:** 3
**Rating:** 4
**Confidence:** 4

**Summary:**

The paper presents a new curriculum learning paradigm for addressing exploration tasks with hard constraints. The gist of the approach is the use of a teacher that can shape the reward of the learner, specifically constructed to reflect a sparse reward structure, so that they can learn with increasingly lower budgets. A theoretical analysis is provided for the case of a binary-tree MDP. Empirical evaluation covers 3 domains (4 if counting both PuddlePoint), with the last one being a real-world LLM solving a math problem using a chain-of-thought step count as a budget. The results show improved performance under strict test-time constraints, including higher success rates and shorter outputs for LLMs, without violating the budget.

**Questions:**

1. What are your thoughts about the theoretical analysis of this approach beyond the binary-tree MDP? Obviously it might be much more challenging to provide sound claims, but I wonder about your intuitions. When will it work well and when will it fail?
2. Have you explored generalization to unseen tasks beyond the ones used during training?
3. How sensitive is the approach to curriculum parameters like the performance threshold beta?

**Ethical Concerns:**

["NO or VERY MINOR ethics concerns only"]

**Final Justification:**

The authors have engaged in the discussion and answered my questions to my satisfaction. Given their responses and the other reviews, I keep my score as is.

**Limitations:**

Yes

**Paper Formatting Concerns:**

No formatting concerns

**Quality:**

3

**Strengths And Weaknesses:**

Strengths:
+ The problem formulation of exploration under hard constraints or a budget is an important step towards practical agents.
+ Using curriculum learning with RL exploration for LLMs is a novel and promising approach with practical potential.
+ The paper is mostly well-written and easy to follow.

Weaknesses:
- While the idea of the curriculum is novel and has merit, the outcome of the proposed approach is limited: Theorem 1 covers only the binary-tree MDP case. Most exploration problems are more complex than that, so it is unclear how much merit this theorem provides. Moreover, the curriculum relies on accurate performance estimation at various constraint levels using only the learner's rollouts. In complex domains like MathLLM, where each rollout is expensive and noisy, this may limit the teacher's ability to adaptively select effective budgets.
- The paper cites a recent survey of curriculum learning for RL agents [19], but ignores the content of the survey beyond the reference. It can provide useful vocabulary for this work (e.g., threshold performance).
- The paper similarly mentions a recent work that is similar [14], but refers to it beyond the reference number only in the empirical methods. This work seems to be the closest to this one, so it should be better explained (in the introduction, not as an afterthought in the methods) what the gaps are that [14] still has and are addressed here.

---

> ### Author Rebuttal · Authors · 2025-07-30
>
> Thank you for carefully reviewing our paper! We greatly appreciate your feedback. Please see below our responses to your comments.
>
> -----
>
> **1. While the idea of the curriculum is novel and has merit, the outcome of the proposed approach is limited: Theorem 1 covers only the binary-tree MDP case. Most exploration problems are more complex than that, so it is unclear how much merit this theorem provides … What are your thoughts about the theoretical analysis of this approach beyond the binary-tree MDP? Obviously it might be much more challenging to provide sound claims, but I wonder about your intuitions. When will it work well and when will it fail?**
>
> We note that binary-tree (or chain-style) environments are canonical settings frequently used in exploration-focused RL theory due to their interpretability and the analytical insights they offer. Our analysis can indeed be extended to more general structures such as multi-branch tree MDPs or star-shaped directed graph MDPs, though the derivations become more involved. For clarity and ease of exposition, we chose to focus on the binary-tree case.
>
> Interestingly, as shown in [1], many reasoning tasks with concrete goals in LLMs can be modeled as star-shaped directed graphs. In this light, our analysis is relevant to more practical problems as well. Overall, the theoretical guarantees provided in this simplified setting help build foundational understanding, while our empirical results validate the method across both RL and LLM domains.
>
>  [1] Bachmann and Nagarajan, 2024. “The Pitfalls of Next-Token Prediction”. ICML 2024
>
> -----
>
> **2. Moreover, the curriculum relies on accurate performance estimation at various constraint levels using only the learner's rollouts. In complex domains like MathLLM, where each rollout is expensive and noisy, this may limit the teacher's ability to adaptively select effective budgets.**
>
> We would like to clarify that our method does not require collecting additional rollouts at different constraint levels to adapt the training budget $\alpha$. Instead, the teacher selects $\alpha$ based solely on the history of rollouts already collected during the student’s training. As a result, there is no added computational overhead. While this may introduce some noise in budget estimation, our experimental results demonstrate that the adaptively selected budgets are still effective in practice. We will clarify this in the implementation section of the revised draft.
>
> -----
>
> **3. The paper similarly mentions a recent work that is similar [14], but refers to it beyond the reference number only in the empirical methods. This work seems to be the closest to this one, so it should be better explained (in the introduction, not as an afterthought in the methods) what the gaps are that [14] still has and are addressed here.**
>
> There are several key conceptual and methodological differences between the ProCuRL-Target framework and our work: (1) ProCuRL-Target focuses on curriculum design for contextual RL, whereas our work targets RL with trajectory-level constraints — Section 4.2 discusses why ProCuRL-Target is not practical for large-scale constrained RL settings; (2) their curriculum strategy is based on the ZPD principle, while ours is inspired by self-paced learning; (3) our theoretical formulation, analysis, and proof techniques are entirely different and novel; and (4) unlike ProCuRL-Target, we include LLM-based experiments in the main paper. We will make these distinctions explicit in the introduction of the final draft.
>
> -----
>
> **4. Have you explored generalization to unseen tasks beyond the ones used during training?**
>
> Yes, in the LLM experiments, performance under test-time constraints is evaluated on unseen test sets for both the SVAMP and GSM8K datasets, demonstrating the model's ability to generalize beyond the training tasks.
>
> -----
>
> **5. How sensitive is the approach to curriculum parameters like the performance threshold beta?**
>
> We would like to emphasize that in our experiments, we did not fine-tune the performance threshold $\beta$. We used a standard value of $\beta = 0.5$, which aligns with the principle of intermediate difficulty, and kept it fixed across all settings. In general, smaller values of $\beta$ allow a faster decrease in the training budget $\alpha$, resulting in quicker convergence to the target constraint. Conversely, larger $\beta$ values slow down this progression.
>
> To evaluate robustness, we conducted an ablation study on $\beta$ across all RL environments. We will include these results in the Appendix and present them below.
>
> **Performance of CuRL-TrajC w.r.t. performance threshold $\beta$: BinaryTree**
>
> | $\beta$ | 20K | 30K | 40K |
> | ----- | --- | --- | --- |
> | 0.1 | $0.24$ | $0.73$ | $0.90$ |
> | 0.2 | $0.29$ | $0.73$ | $0.90$ |
> | 0.3 | $0.30$ | $0.74$ | $0.90$ |
> | 0.4 | $0.30$ | $0.73$ | $0.90$ |
> | 0.5 (paper) | $0.23$ | $0.70$ | $0.90$ |
> | 0.6 | $0.17$ | $0.60$ | $0.86$ |
> | 0.7 | $0.10$ | $0.41$ | $0.82$ |
> | 0.8 | $0.04$ | $0.23$ | $0.56$ |
> | 0.9 | $0.01$ | $0.05$ | $0.16$ |
> |
>
> **Performance of CuRL-TrajC w.r.t. performance threshold $\beta$: PuddleGrid-Single**
>
> | $\beta$ | 2K | 3K | 4K | 5K |
> | ----- | --- | --- | --- | --- |
> | 0.1 | $0.01$ | $0.13$ | $0.55$ |  $0.93$ |
> | 0.2 | $0.00$ | $0.35$ | $0.71$ |  $0.95$ |
> | 0.3 | $0.06$ | $0.38$ | $0.85$ |  $0.97$ |
> | 0.4 | $0.07$ | $0.37$ | $0.80$ | $0.97$ |
> | 0.5 (paper) | $0.10$ | $0.33$ | $0.77$ | $0.97$ |
> | 0.6 | $0.07$ | $0.28$ | $0.77$ |  $0.97$ |
> | 0.7 | $0.05$ | $0.19$ | $0.65$ |  $0.96$ |
> | 0.8 | $0.07$ | $0.16$ | $0.54$ |  $0.96$ |
> | 0.9 | $0.04$ | $0.08$ | $0.22$ | $0.60$ |
> |
>
> **Performance of CuRL-TrajC w.r.t. performance threshold $\beta$: PuddleGrid-Multi**
>
> | $\beta$ | 5K | 10K | 15K |
> | ----- | --- | --- | --- |
> | 0.1 | $0.44$ | $0.91$ | $0.98$ |
> | 0.2 | $0.37$ | $0.98$ | $0.99$ |
> | 0.3 | $0.40$ | $0.91$ | $0.98$ |
> | 0.4 | $0.30$ | $0.94$ | $0.97$ |
> | 0.5 (paper) | $0.30$ | $0.94$ | $0.98$ |
> | 0.6 | $0.26$ | $0.73$ | $0.94$ |
> | 0.7 | $0.30$ | $0.81$ | $0.98$ |
> | 0.8 | $0.27$ | $0.56$ | $0.81$ |
> | 0.9 | $0.20$ | $0.40$ | $0.42$ |
> |
>
> Note that $\beta=0$ and $\beta=1$ correspond to the Target and Unconstrained baselines respectively, and are therefore excluded from the tables.
>
> -----
>
> **6. The paper cites a recent survey of curriculum learning for RL agents [19], but ignores the content of the survey beyond the reference. It can provide useful vocabulary for this work (e.g., threshold performance).**
>
> We thank the reviewer for this helpful suggestion and will take it into consideration. Throughout the paper, we have used standard terminology from the curriculum learning and reinforcement learning literature.
>
> -----
>
> We hope that our responses can address your concerns and are helpful in improving your rating. If you have any other comments or feedback, please let us know! We are looking forward to hearing back from you! Thank you again for the review.

---

### Official Review · Reviewer_sU2F · 2025-07-03

**Clarity:** 3
**Significance:** 3
**Originality:** 3
**Rating:** 4
**Confidence:** 3

**Summary:**

- In the paper the authors try to address the problem of deployment constraints in terms of budget or safety requirements while training RL agents.
- The authors propose a curriculum learning strategy to dynamically/adaptively adjust the constraints (from relaxed to strict) during the training of RL agents based on their performance.
- The authors try to find answers to the question: “How to guide the training and get the best performance when deployment-time constraints advance”
- The paper introduces a curriculum learning framework to have a sample and compute efficient training strategy under strict trajectory-level constraints so that the final policy can adapt to strict resource and safety requirements.
- The authors test this framework on a toy binary-tree MDP example and as well as multi-task navigation and LLM based math reasoning tasks.
- When applying this curriculum learning based strategy for LLM tasks, they claim to have compressed output tokens (a deployment constraint) with a 4.5x inference speedup on consumer hardware.
- The paper formulates this strategy as a constrained MDP problem where the RL agent (student) tries to roll out the policy to accomplish a task (from the task space) given a required constrained/budget, get the training-time (constrained) rewards and updates the policy. A teacher component is responsible for adjusting/replacing the deployment-time budget with the relaxed training-time cost.
  - This is done to ensure that the  final performance of the policy is eps-near-optimal.
  - The framework now tries to optimize the teacher component to select permissible training time budget so that the required policy performance is achieved efficiently in terms of cost and sample complexity.

**Questions:**

- How can we impose other real-time constraints like memory constraints in terms of model size, quantization, GPU memory, etc?
- Do we have any numbers/plots in terms of latency or inference-speed?

Suggestions:
- Models that can be deployed under different/dynamic test-time constraints or budgets can be trained (similar to [Matryoshka style learning](https://arxiv.org/abs/2205.13147)) and can be adaptively used as per budget requirements as they change with time.

**Ethical Concerns:**

["NO or VERY MINOR ethics concerns only"]

**Final Justification:**

The authors tried to resolve some of the concerns during the rebuttal.
- Authors mentioned the work is not directly related to ProCurL-Target framework.
- Additional LLM-based experiments were conducted that could be more suitable given the proposals and title of the paper.
- Authors mentioned that system-level constraints like memory usage, quantization, or model size fall outside the current scope of o the proposed framework.

There are still some future directions that were agreed upon by the authors in the rebuttal and those would be great if worked upon or at least included in the future directions in the final version of the paper.

This work is a good combination of research and practical engineering. I hope to have a better and more comprehensive final version of the paper. I would still like to maintain my score.

**Limitations:**

- The authors mentioned extending this work to the programming/code generation domains as well rather than just for math.
- How the model optimizes for the given constraints during the training is still unclear.

**Paper Formatting Concerns:**

- It would be nice to define what is meant by “Performance” in the plots or how it is calculated.

**Quality:**

3

**Strengths And Weaknesses:**

** Strengths:**
- The authors try to solve the sparse reward problem of training with the strict deployment constraints from the beginning and instead try to introduce a curriculum strategy.
- The paper did a good job in providing a thorough theoretical analysis of how the proposed strategy would work for the toy example of binary-tree MDP.
  - The paper is able to prove that imposing a target/deployment-time budget at the start of the training would require an exponential number of rollouts as compared to polynomial $O(H^3)$ sample complexity in case of curriculum strategy.
  - The paper shows theoretically that the proposed strategy reduces the number of rollouts required to learn an eps-suboptimal policy.
- Results from experiments on other environments (navigation and LLM reasoning) supported the theoretical results.
The results and the plots are nicely presented.

**Weakness:**
- It would be nice to have some tabular results as well to have some numerical analysis.

- While there have been good efforts in terms of having theoretical and empirical analysis, this work seems to be deriving few things from the ProCurl-Target work (it does not mean that this fact downweighs this paper’s efforts), this can still be experimented and extended to some other high-dimensional tasks and other LLM or VLM domains to have more real-world constraints.

- This work can present more LLM based use-cases as well given the title of the paper.
- It would be can be worth trying to incorporate some constraints related to:
  - different (reduced) input token numbers
  - token generation: tokens/sec
  - other test-time scaling constraints.
  - some subjective constraints like safety, preference, etc.

---

> ### Author Rebuttal · Authors · 2025-07-30
>
> Thank you for carefully reviewing our paper! We greatly appreciate your feedback. Please see below our responses to your comments.
>
> -----
>
> **1. It would be nice to have some tabular results as well to have some numerical analysis.**
>
> We focus on test performance curves and training plots to highlight the learning dynamics and how agent behavior evolves throughout training. We thank the reviewer for the suggestion and are happy to include tabular results in the Appendix, reporting performance metrics along with a brief analysis.
>
> -----
>
> **2. While there have been good efforts in terms of having theoretical and empirical analysis, this work seems to be deriving few things from the ProCurl-Target work (it does not mean that this fact downweighs this paper’s efforts), this can still be experimented and extended to some other high-dimensional tasks and other LLM or VLM domains to have more real-world constraints.**
>
> We would like to clarify that our work does not derive techniques from the ProCuRL-Target framework. There are several key conceptual and methodological differences: (1) ProCuRL-Target focuses on curriculum design for contextual RL, whereas our work targets RL with trajectory-level constraints — Section 4.2 discusses why ProCuRL-Target is not practical for large-scale constrained RL settings; (2) their curriculum strategy is based on the ZPD principle, while ours is inspired by self-paced learning; (3) our theoretical formulation, analysis, and proof techniques are entirely different and novel; and (4) unlike ProCuRL-Target, we include LLM-based experiments in the main paper and have added further LLM results in this rebuttal.
>
> -----
>
> **3. This work can present more LLM-based use-cases as well, given the title of the paper.**
>
> We have conducted additional LLM-based experiments, which will be included in the revised version of the paper.
>
> The new experiments include:
>
> - A model of different family and size — Qwen2.5-Math-1.5B — evaluated on both datasets.
>
> - A variable target setting, where each task $t$ has its own target budget $\alpha^*$, determined based on the original model response length. This setting is applied across both models and datasets, resulting in four additional experiments.
>
> We present the results in tabular form below.
>
> **Experiments with Qwen2.5-Math-1.5B: SVAMP**
>
> | Method | 50K | 100K | 150K | 200K | 250K |
> | ----- | --- | --- | --- | --- | --- |
> | CuRL-TrajC (ours) | $0.59$ | $0.78$ | $0.81$ | $0.80$ | $0.81$ |
> | ExpSchedule | $0.00$  | $0.00$ | $0.00$ |  $0.00$ | $0.64$ |
> | IID | $0.00$  | $0.00$ | $0.02$ | $0.20$ | $0.29$ |
> | Unconstrained | $0.00$ | $0.00$ | $0.00$ | $0.00$ | $0.00$ |
> | Target | $0.00$ | $0.00$ | $0.57$ | $0.61$ | $0.60$ |
> |
>
> **Experiments with Qwen2.5-Math-1.5B: GSM8K**
>
> | Method | 200K | 300K | 400K | 500K |
> | ----- | --- | --- | --- | --- |
> | CuRL-TrajC (ours) | $0.37$ | $0.56$ | $0.58$ | $0.62$ |
> | ExpSchedule | $0.00$  | $0.00$ | $0.04$ |  $0.50$ |
> | IID | $0.02$  | $0.03$ | $0.13$ | $0.17$ |
> | Unconstrained | $0.00$ | $0.00$ | $0.00$ | $0.00$ |
> | Target | $0.23$ | $0.25$ | $0.25$ | $0.25$ |
> |
>
> **Experiments with variable targets Qwen2.5-Math-1.5B: SVAMP**
>
> | Method | 50K | 100K | 150K | 200K | 250K |
> | ----- | --- | --- | --- | --- | --- |
> | CuRL-TrajC (ours) | $0.27$ | $0.76$ | $0.77$ | $0.77$ | $0.77$ |
> | ExpSchedule | $0.00$  | $0.00$ | $0.00$ |  $0.00$ | $0.60$ |
> | IID | $0.00$  | $0.00$ | $0.01$ | $0.02$ | $0.07$ |
> | Unconstrained | $0.00$ | $0.00$ | $0.00$ | $0.00$ | $0.00$ |
> | Target | $0.00$ | $0.00$ | $0.00$ | $0.00$ | $0.00$ |
> |
>
> **Experiments with variable targets Qwen2.5-Math-1.5B: GSM8K**
>
> | Method | 200K | 300K | 400K | 500K |
> | ----- | --- | --- | --- | --- |
> | CuRL-TrajC (ours) | $0.26$ | $0.49$ | $0.56$ | $0.61$ |
> | ExpSchedule | $0.00$  | $0.00$ | $0.01$ |  $0.23$ |
> | IID | $0.01$  | $0.02$ | $0.08$ | $0.05$ |
> | Unconstrained | $0.00$ | $0.00$ | $0.00$ | $0.00$ |
> | Target | $0.00$ | $0.00$ | $0.00$ | $0.00$ |
> |
>
> **Experiments with variable targets MetaMath: SVAMP**
>
> | Method | 50K | 100K | 150K | 200K | 250K |
> | ----- | --- | --- | --- | --- | --- |
> | CuRL-TrajC (ours) | $0.06$ | $0.46$ | $0.61$ | $0.63$ | $0.63$ |
> | ExpSchedule | $0.00$  | $0.00$ | $0.00$ |  $0.04$ | $0.58$ |
> | IID | $0.00$  | $0.00$ | $0.00$ | $0.01$ | $0.00$ |
> | Unconstrained | $0.00$ | $0.00$ | $0.00$ | $0.00$ | $0.00$ |
> | Target | $0.00$ | $0.00$ | $0.00$ | $0.00$ | $0.00$ |
> |
>
> **Experiments with variable targets MetaMath: GSM8K**
>
> | Method | 200K | 300K | 400K | 500K |
> | ----- | --- | --- | --- | --- |
> | CuRL-TrajC (ours) | $0.18$ | $0.38$ | $0.41$ | $0.42$ |
> | ExpSchedule | $0.00$  | $0.00$ | $0.00$ |  $0.27$ |
> | IID | $0.00$  | $0.00$ | $0.00$ | $0.00$ |
> | Unconstrained | $0.00$ | $0.00$ | $0.00$ | $0.00$ |
> | Target | $0.00$ | $0.00$ | $0.00$ | $0.00$ |
> |
>
> -----
>
> **4. It would be worth trying to incorporate some constraints related to: i) different (reduced) input token numbers, ii) token generation: tokens/sec, iii) other test-time scaling constraints, iv) some subjective constraints like safety, preference, etc., v) memory constraints in terms of model size, quantization, GPU memory.**
>
> We would like to note that our problem setup — and the proposed curriculum strategy — applies broadly to scenarios where trajectory-level constraints can be expressed via a cost function over the output or response trajectories. More concretely:
>
> - Reaching a goal under navigation cost constraints, as demonstrated in the paper through the agent's path cost in MiniGrid.
>
> - LLM output generation with reduced reasoning tokens, also covered in the paper as trajectory length constraints.
>
> - In the rebuttal, we present additional LLM experiments involving variable token length constraints based on original model response length.
>
> Beyond these, our method naturally extends to:
>
> - Harmfulness constraints in LLMs, where a harmfulness or toxicity score can serve as the cost function.
>
> - Code generation tasks, where generated programs must satisfy test cases or meet resource/memory constraints — both of which can be captured via cost functions over the generated output.
>
> While system-level constraints like memory usage, quantization, or model size are important, they fall outside the current scope of our framework, which is designed to handle constraints that can be expressed as cost functions over output trajectories.
>
> In general, any constraint on response trajectories that can be formalized as a cost function is compatible with our framework. While it’s not practical to include all possible scenarios in the paper, we aimed to showcase a representative set of impactful applications.
>
> -----
>
> **5. Do we have any numbers/plots in terms of latency or inference speed?**
>
> At the end of Section 4.3, we provide a qualitative comparison of the inference time comparison between the model fine-tuned using our approach and the original model to highlight its practical impact. In the revised version, we are happy to include a more detailed comparison to further support this analysis.
>
> -----
>
> **6. The authors mentioned extending this work to the programming/code generation domains as well rather than just for math.**
>
> We focused on mathematical tasks because they are widely regarded as standard benchmarks for evaluating the reasoning capabilities of LLMs. Nevertheless, as noted in our earlier response, while math problems often use reasoning length as a constraint, code generation introduces different types of constraints. In addition to achieving the task objective, generated code may need to meet specific security requirements, satisfy test cases, or stay within memory or compute resource limits during execution. These can be naturally incorporated into our framework by defining appropriate cost functions — e.g., test-case pass rates or programmatic evaluations — over the code/program space.
>
> -----
>
> **7. How the model optimizes for the given constraints during the training is still unclear.**
>
> The student model is optimized using a standard RL update rule, where the reward function depends on the training constraint or budget selected by the teacher at each step t. As a result, the reward function is non-stationary throughout training, i.e., starting with an unconstrained reward setting and gradually shifting toward the target constraint. The student thus learns under a progressively constrained reward landscape. We will clarify this point in the revised draft.
>
> -----
>
> **8. Models that can be deployed under different/dynamic test-time constraints or budgets can be trained (similar to Matryoshka style learning) and can be adaptively used as per budget requirements as they change with time.**
>
> This is a very interesting suggestion, and we thank the reviewer for highlighting it. We view this as a promising direction for future work.
>
> -----
>
> **9. ​​It would be nice to define what is meant by “Performance” in the plots or how it is calculated.**
>
> Performance refers to the agent’s average return when evaluated on the test tasks under the target test-time constraints. We will clarify this in the revised version.
>
> -----
>
> We hope that our responses can address your concerns and are helpful in improving your rating. If you have any other comments or feedback, please let us know! We are looking forward to hearing back from you! Thank you again for the review.

---

> > ### Author Response · Authors · 2025-08-06
> >
> > As the discussion period is ending soon, we are writing to thank the reviewer for their constructive feedback. We hope that our responses have addressed your concerns and are helpful in improving your rating. We will incorporate the reviewer's feedback and our responses in the updated paper.

---

> ### Comment · Area_Chair_oJ7t · 2025-08-06
>
> Please remember to respond to authors' rebuttal as soon as possible.
>
> Thank you!
>
> -AC

---

> ### Comment · Reviewer_sU2F · 2025-08-07
>
> Thanking the authors for addressing the questions.
> I went through the comments and discussions from other reviewers as well and it seems that the authors have tried to carefully address their concerns - some of which were shared across other reviewers.
> The authors did a good job in putting together some more experiments and ablations that would make the contributions more clear.
> It would be nice to include more description/comparison to reference mentioned in the rebuttal comments.
>
> There are still some future directions that were agreed upon by the authors in the rebuttal and those would be great if worked upon or at least included in the future directions in the final version of the paper.
>
> Appreciation for the authors was putting efforts in this direction and releasing their work. This work is a good combination of research and practical engineering. I hope to have a better and more comprehensive final version of the paper. I would still like to maintain my score.
>
> Best wishes! :)

---

> > ### Author Response · Authors · 2025-08-08
> >
> > We would like to thank the reviewer for their constructive feedback and engagement during the discussion period! We will incorporate the reviewer's feedback and our responses (including additional experiments and comparison with existing methods). We will also expand on the future work directions in the paper. We sincerely appreciate the reviewer's input in helping us improve the paper!

---

### Decision · Program_Chairs · 2025-09-17

**Decision:**

Accept (poster)

**Comment:**

The authors present a framework for introducing a curriculum to gradually incorporate constraints into constrained reinforcement learning settings with demonstrations of its use in a variety of settings including binary-tree MDPs and math reasoning tasks using LLMs. Additionally, they provide some theoretical analysis to motivate why this curriculum learning approach can accelerate training.

Strengths: The work is theoretically grounded and the demonstrations of the method in both very simple and very complex settings is impressive.  Using curriculum learning with RL exploration for LLMs is a novel and promising approach with practical potential. Reviewers also found the paper easy to follow.

Weaknesses: There are similarities to prior work ProCurl-Target. The theoretical analysis is only on the binary-tree MDP case. While there are case studies on toy MDPs and math reasoning tasks, there is nothing inbetween on more standard RL benchmarks. Evaluation was only performed on a few types of constraints and with only very basic methods as baselines.

Reasons for decision: Overall the contribution of this work is impressive both empirically and theoretically. There are some weaknesses, i.e. the theory is only relevant for the binary-tree MDP case and the work would benefit from filling in the gap of having experiments on other RL domains with more varied forms of constraints.

Rebuttal summary: The reviewers had questions about the sensitivity of this method to hyperparameters, as well as questions about other types of constraints and use cases and methods that were compared against. The authors provided additional results with hyperparameter sweeps, more baseline comparisons, and demonstrations of more LLM use cases.